



# Deriving Arctic 2 m air temperatures over snow and ice from satellite surface temperature measurements

Pia Nielsen-Englyst[1,2], Jacob L. Høyer[2], Kristine S. Madsen[2], Rasmus T. Tonboe[2] and Gorm Dybkjær[2]

[1] Technical University of Denmark (DTU), DK-2800 Kongens Lyngby, Denmark
[2] Danish Meteorological Institute (DMI), DK-2100 Copenhagen Ø, Denmark

*Correspondence to*: Pia Nielsen-Englyst (pne@dmi.dk)

**Abstract.**

The Arctic region is responding heavily to climate change, and yet, the air temperature of Arctic, ice covered areas is heavily under-sampled when it comes to in situ measurements, and large uncertainties exist in weather- and reanalysis products. This
paper presents a method for estimating daily mean 2 meter air temperatures (T2m) in the Arctic from satellite observations of skin temperature, using the Arctic and Antarctic ice Surface Temperatures from thermal Infrared (AASTI) satellite dataset, providing spatially detailed observations of the Arctic. The method is based on a linear regression model which has been developed using in situ observations and daily mean satellite ice surface skin temperatures combined with a seasonal variation to estimate daily T2m. The satellite derived T2m product including estimated uncertainties covers clear sky snow
and ice surfaces in the Arctic region during the period 2000-2009. Comparison with independent in situ measured T2m gives average correlations of 95.5% and 96.5% and average root mean square errors of 3.47°C and 3.20°C for land ice and sea ice, respectively. The reconstruction provides a much better spatial coverage than the sparse in situ observations of T2m in the Arctic, is independent of numerical weather prediction model input and it therefore provides an important alternative to simulated air temperatures to be used for assimilation or global surface temperature reconstructions. A comparison between
in situ T2m versus T2m from satellite and ERA-Interim shows that the T2m derived from satellite observations validate similar or better than ERA-Interim estimates in the Arctic.

## 1 Introduction

The Arctic climate is changing rapidly with surface temperatures rising faster than other regions of the world due to Arctic amplification (Graversen et al., 2008; Pithan and Mauritsen, 2014). Meteorological measurements show that the 2000s were
the warmest decade in Greenland since meteorological measurements started in the 1780s (Box et al., 2019; Cappelen, 2016; Masson-Delmotte et al., 2012). The Arctic surface air temperature is an important indicator of regional and global climate change as most simulations by global climate models indicate that any warming in the global climate will be amplified at the northern high latitudes (e.g. Holland and Bitz, 2003; Overland et al., 2018). Traditionally, near surface air temperatures have been measured at the height of 1-2 m using automatic weather stations (AWSs) or buoys (Hansen et al., 2010; Jones et al.,



2012; Rayner, 2003). Extreme temperatures, winds and the remoteness of the Arctic make in situ observations in the Arctic temporally and spatially sparse (Reeves Eyre and Zeng, 2017), and challenging. In particular, it is difficult to achieve climate-quality temperature records for this region. The most practical way to get continuous and spatially broad measurements of the data-sparse Arctic is through satellite remote sensing. However, satellites with infrared sensors in the

atmospheric window region of 10-12 micron wavelength measure the ice surface skin temperature ($IST_{skin}$) during clear skies whereas the current global temperature products estimate the near surface air temperature as are measured AWSs and buoys. The standard measurement height is 2 m, but it varies with snow depth at the sites (World Meteorological Organization, 2014). However, all measurements are relatively near the surface and are therefore also often called "surface air temperatures". The surface skin temperature may differ considerably from the surface air temperature during melting

conditions, but during other conditions the skin and surface air temperature may be more or less the same (Nielsen-Englyst et al., 2019).

To benefit from the good coverage of satellite surface temperature data, we have explored the relationships between the surface air temperature and the satellite measurements. Several studies have compared satellite retrieved $IST_{skin}$ and T2m from AWSs located on the Greenland Ice Sheet (GrIS; Dybkjær et al., 2012a; Hall et al., 2008, 2012; Koenig and Hall, 2010;

Shuman et al., 2014) and over the Arctic sea ice (Dybkjær et al., 2012) and found temperature differences of which a significant part could be attributed to the temperature difference between T2m and $IST_{skin}$. Previously, work has been done to investigate the relationship between the surface and near-surface air temperature over ice using in situ observations (Adolph et al., 2018; Hall et al., 2008, 2004; Hudson and Brandt, 2005; Nielsen-Englyst et al., 2019; Vihma et al., 2008). Nielsen-Englyst et al. (2019) found that on average T2m is 0.65-2.65°C warmer than $IST_{skin}$ with variations depending on location of

the measurement i.e. in the lower ablation zone, upper-middle ablation zone, accumulation zone, seasonal snow cover and sea ice. The T2m-$IST_{skin}$ difference was found to vary with season with smallest differences around noon and early afternoon during spring, fall and summer during non-melting conditions. Furthermore, wind speed and cloud cover were identified as key parameters determining the T2m- $IST_{skin}$ difference.

Given the difficulties of operating equipment in the harsh Arctic conditions, the potential for using satellite $IST_{skin}$ to

estimate T2m is large in this region. The greatest limitation of satellite-derived infrared temperatures is cloud cover. Hence, a satellite-derived, clear-sky, surface temperature record can be significant different from an all-sky surface temperature record (Koenig and Hall, 2010; Nielsen-Englyst et al., 2019).

This work, starting with (Nielsen-Englyst et al., 2019), has been initiated to estimate T2m from satellite observations covering the snow and ice covered parts of the Arctic, in order to provide spatially-detailed observations for the areas

unobserved by in situ stations and to supplement the in situ observations already available. The investigation of the $IST_{skin}$ versus T2m relationships over ice and the derivation of T2m from $IST_{skin}$ were also identified in Merchant et al. (2013) as important areas for improving the understanding of the surface temperature of the Earth.

In this paper a regression-based approach has been used to estimate daily T2m using satellite $IST_{skin}$ and a seasonal cycle function as predictors, based upon the work presented in Høyer et al. (2018). Similar efforts have been done to estimate near





surface air temperatures over land, ocean and lakes using satellite observations (Good, 2015; Good et al., 2017; Høyer et al., 2018). The previous work has mostly been done as a part of the European Union's Horizon2020 project EUSTACE (EU Surface Temperatures for All Corners of Earth, 2015-2019, https://www.eustaceproject.org), with the aim to produce a globally complete daily near surface temperature analysis since 1850 using a combination of satellite and in situ observations.

This paper is structured such that Sect. 2 describes the in situ data and the satellite data. Section 3 presents the method used to estimate daily T2m and uncertainties. The resulting T2m dataset and its validation are presented in Sect. 4 and discussed in Sect. 5. Conclusions are given in Sect. 6.

## 2 Data

### 2.1 In Situ data

In situ observations of near surface air temperatures have been collected from weather stations, expeditions and campaigns covering ice and snow surfaces to assemble the DMI-EUSTACE database. The database includes quality controlled and uniformly formatted temperature observations covering ice and snow surfaces, during 2000-2009 (Høyer et al., 2018). Over Arctic land ice/snow we use the Programme for Monitoring of the Greenland Ice Sheet (PROMICE) data provided by the Geological Survey of Denmark and Greenland (GEUS; Ahlstrøm et al., 2008), the Atmospheric Radiation Measurement (ARM) Program data from the North Slope of Alaska (Ackerman and Stokes, 2003; Stamnes et al., 1999), and the Greenland Climate Network data (GC-Net; Kindig, 2010; Shuman et al., 2001; Steffen and Box, 2001). Only PROMICE data from the middle-upper ablation zone and accumulation zone have been used to ensure that data are only acquired over permanently snow or ice covered surfaces. Data on Arctic sea ice are primarily retrieved from the meteorological observation archive at the European Centre for Medium-Range Weather Forecasts (ECMWF) MARS data storage facility, providing 196 unique data series from drifting buoys of which 11 measure $IST_{skin}$ as well. These sea ice data are supplemented with data from 10 U.S. Army Cold Regions Research Engineering Laboratory (CRREL) mass balance buoys (Perovich et al., 2016; Richter-Menge et al., 2006) and observations from the research vessel, POLARSTERN, operated by the Alfred-Wegener-Institute, operating the sea ice covered parts of the Arctic Ocean (Knust, 2017). We also use air temperature measurements obtained from ice buoys deployed in the Fram Strait region within the framework of the Fram Strait Cyclones (FRAMZY) campaigns during the years 2002, 2007, and 2008 as well as air temperatures from the Arctic Climate System Study (ACSYS) campaign in 2003 (Brümmer et al., 2011b, 2011c, 2012b, 2012a). Finally, we use data from two ice buoy campaigns operated by the Meteorological Institute of the University of Hamburg within the framework of the integrated EU research project DAMOCLES (Developing Arctic Modelling and Observing Capabilities for Long-term Environmental Studies; Brümmer et al., 2011a).

The different in situ types measure the air temperature at different heights that furthermore differ over time depending on the amount of snow fall, snow drift and snow melt. Here, we will refer to T2m for all observation types regardless of these





variations. Nielsen-Englyst et al. (2019) showed small changes (<0.22°C) in T2m-IST$_{skin}$ differences when using only observations within the measurement range of 1.90-2.10 m in height compared to using all measurements (ranging in measurement height from 0.3 m to 3 m). The accuracy of the air temperature sensors for all observation sites is approximated to 0.1°C (Hall et al., 2008; Høyer et al., 2017b). Few data sources provide both skin and air temperatures e.g. the PROMICE

and ARM stations over land ice and 11 of the buoys from ECMWF operational data stream over sea ice. The PROMICE skin temperatures have been calculated from up-welling longwave radiation, measured by Kipp & Zonen CNR1 or CNR4 radiometer, assuming a surface longwave emissivity of 0.97 (van As, 2011). The 11 buoy time series from ECMWF that provide IST$_{skin}$ observations as well are likely not measuring the actual skin temperature. The buoy thermistor is placed on the underside of the buoy which is set out on top of the ice. The buoys typically get buried in snow and the measured

temperature becomes an internal snow temperature. However, in this analysis the buoy temperature measurements will be treated and counted as IST$_{skin}$ measurements, as we have no information on the snow depth on top of the buoys. All in situ data have been screened for spikes and other unrealistic data artefacts by visual inspection. Afterwards, the in situ observations have been averaged to daily temperatures using all available observations. Figure 1 shows the number of daily averaged in situ observations each year during 2000-2009 of IST$_{skin}$ and T2m over Arctic land ice and sea ice, respectively.

In total 65,810 observations with daily T2m and 7,681 observations with daily IST$_{skin}$ are available. However, only 624 of these cover Arctic sea ice. See Table 1 for more information on the in situ observations used in this study.

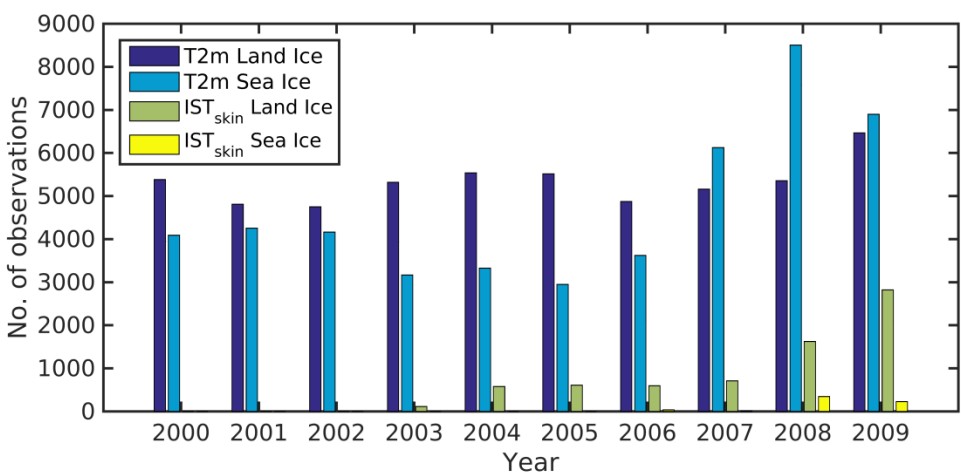

**Figure 1: Total number of daily averaged in situ observations of T2m and IST$_{skin}$ over Arctic land ice and sea ice per year covering the period 2000-2009.**

**Table 1: Overview of in situ observations used in this study, covering 2000-2009.**

| No. of sites, (AWS, buoys or ships) | No. of days with observations | Surface Type | Observation Type | Temperature measurements |
|---|---|---|---|---|



| ACSYS | 7 | 280 | Sea ice | Buoy | T2m |
|---|---|---|---|---|---|
| ARM | 2 | 2,846 | Land snow | AWS | T2m, $IST_{skin}$ |
| CRREL | 10 | 1,031 | Sea ice | Buoy | T2m |
| DAMOCLES | 25 | 2,160 | Sea ice | Buoy | T2m |
| ECMWF | 196 | 27,235 | Sea ice | Buoy | T2m, IST (11 buoys) |
| FRAMZY | 11 | 251 | Sea ice | Buoy | T2m |
| GC-NET | 15 | 29,133 | Land ice | AWS | T2m |
| POLARSTERN | 1 | 189 | Sea ice | Ship | T2m |
| PROMICE | 8 | 2,685 | Land ice | AWS | T2m, $IST_{skin}$ |

## 2.2 Satellite data

The satellite data used in this study is from the Arctic and Antarctic Ice Surface Temperatures from thermal Infrared satellite sensors (AASTI; Dybkjaer et al., 2018; Dybkjær et al., 2014; Høyer et al., 2019) data set, covering high latitude seas, sea ice,

and ice cap surface temperatures based on satellite infrared measurements from the CLARA-A1 data set compiled by EUMETSAT's Climate Monitoring, Satellite Application Facility (Karlsson et al., 2013). The data set is based on one of the longest existing satellite records from the Advanced Very High Resolution Radiometer (AVHRR) instruments on board a long series of NOAA satellites. AASTI contains Level 2 (L2) ice surface skin temperature ($IST_{skin\_L2}$) data processed and error corrected on the original Global Area Coverage (GAC) grid. The first version of the AASTI product, used in this study,

is available from 2000 to 2009 in ~0.05 arc degree resolution. Since 2000, seven different AVHRR instruments have been orbiting the globe, each 14 times per day, and thus providing approximately bi-hourly coverage of the Polar Regions (Figure 2). The number of operational satellites has increased from 2 to 6 from 2000 to 2009.

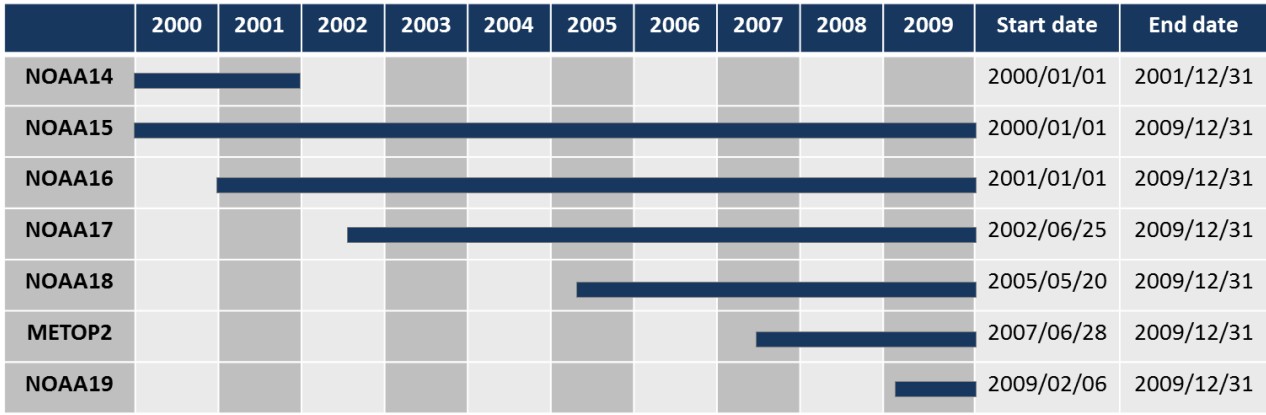

**Figure 2: NOAA and Metop satellites carrying the AVHRR sensor, used for AASTI version 1.**

As discussed in Merchant et al. (2017), satellite-based climate data records should include uncertainty estimates. The AASTI $IST_{skin\_L2}$ data come with uncertainties divided into three independent uncertainty components, each with different



characteristics: the random uncertainty ($\mu_{rnd\_L2}$), a locally systematic uncertainty ($\mu_{local\_L2}$) and a large-scale systematic ("global") uncertainty ($\mu_{glob\_L2}$). These three components have been chosen since they behave differently when aggregating the observations in time or space (see Sect. 3.2). This uncertainty methodology has been developed within the SST community (Bulgin et al., 2016; Rayner et al., 2015) and will be followed here. The total uncertainty on the IST$_{skin\_L2}$,

$\mu_{total\_L2}$ , is calculated by summing each component in quadrature (i.e., square root of sum of squares). Excluding the cloud mask uncertainty, grid-cell systematic uncertainties ($\mu_{glob\_L2}$) are set to a fixed value of 0.1°C to represent systematic uncertainties in the forward models (see e.g. Merchant et al., 1999; Merchant and Le Borgne, 2004). The AASTI IST$_{skin\_L2}$ data also come with a quality level (QL) from 1 (bad data) to 5 (best quality), with the addition of level 0 (no data) (GHRSST Science Team, 2010).

Here, we have aggregated the AASTI IST$_{skin\_L2}$ observations into daily, gridded Level 3 (L3) averages (IST$_{skin\_L3}$) of IST$_{skin\_L2}$ on a fixed 0.25 by 0.25 degrees regular geographical grid. This has been done to facilitate the development of the relationship model and to ease the user uptake. The data in the aggregated files contain mean surface temperature observations from 00 to 24 hours local solar time, but also 3-hourly bin averages of surface temperatures and the number of observations in the eight time bins during each day. The 3-hourly numbers of observations were used for estimating the

satellite sampling throughout the day, and the 3-hourly temperature data to gain confidence in the daily cycle estimates. In the aggregation, all satellite observations with a quality flag of 4 (good) or 5 (best) were used. Figure 3 shows the mean number of observations per day in each of the eight time intervals given in local time for the Arctic region. The variation in the coverage throughout the day is a combined effect of the satellite overpassing and the cloud free conditions during the day. In addition, the fixed 0.25 degrees regular geographical grid results in a decreasing L3 bin area when approaching the

North Pole. The maximum in satellite coverage is generally seen around 80°N with a minimum at the North Pole. Cloud free conditions over the GrIS are primarily observed around noon and early afternoon



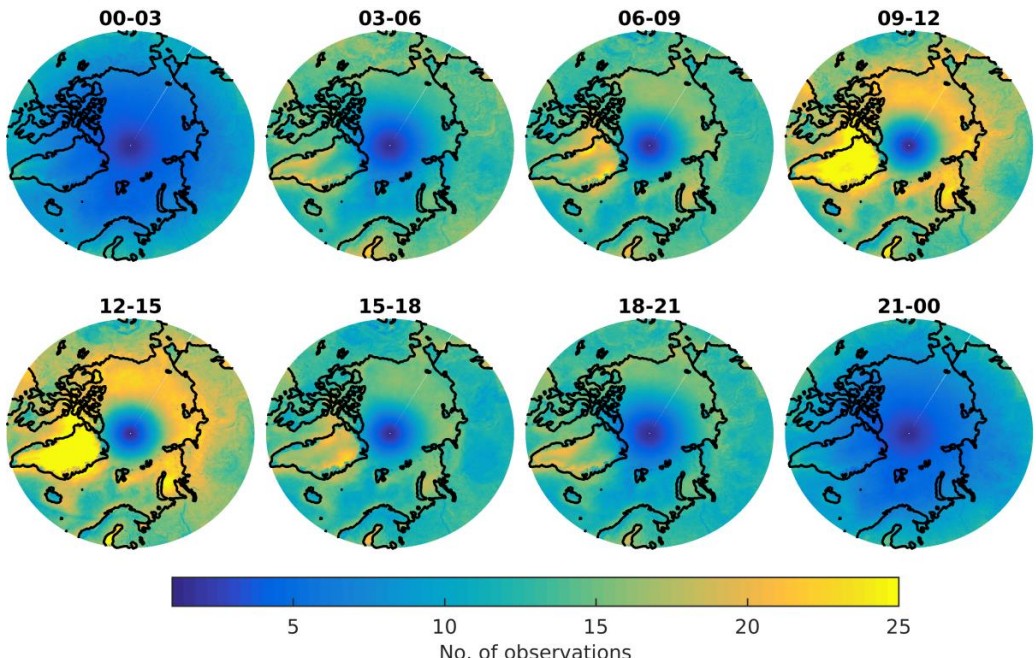

**Figure 3: Mean number of observations per day in the L3 bins for each of the eight local solar time intervals, averaged for the period 2000-2009.**

In order to best resolve the diurnal cycle with satellite information we require data during both night (18-6 local solar time) and day (6-18 local solar time) in order to calculate $IST_{skin\_L3}$. The $IST_{skin\_L3}$ is calculated by averaging all available $IST_{skin\_L2}$ observations for a given date. A few more checks have been set up in order to minimize the temporal sampling errors and the effects of undetected clouds and outliers. Following Høyer et al. (2018), the $IST_{skin\_L3}$ is discarded if one of the following criteria is met:

- $IST_{skin\_L3}$ exceeds +5°C, indicating clear melting conditions or obviously wrong observations.
- The standard deviation of satellite $IST_{skin\_L2}$ during one day exceeds 7.07°C, corresponding to a sinusoidal daily cycle with a difference between day and night of 20°C.
- The difference between $IST_{skin\_L3}$ and the average of all available 3 h bin averages exceeds 10°C.
- $IST_{skin\_L3}$ is more than 10°C colder than the corresponding average of up to 24 neighbouring cloud free observations (in a 5 by 5 grid cell square) with the same surface type.

The criteria above have been derived from analysis and inspection of the satellite data and with considerations to the results presented in Nielsen-Englyst et al. (2019). The satellite-derived $IST_{skin\_L3}$ has seasonal differences in daily variability, with largest standard deviations during summer in Greenland and during winter for sea ice, where the freeze-up of sea ice causes higher variability along the sea ice margin (Fig. 4). The main uncertainty components of the $IST_{skin\_L3}$ estimates are erroneous cloud screening and the spatial variance of snow and ice surface emissivity, which are not accounted for in the retrieval algorithm. The presence of non-detected clouds will contribute to increased standard deviations and usually a cold



$IST_{skin\_L3}$ bias, since the cloud tops and other atmospheric constituents generally are colder than the surface (Dybkjær et al., 2012).

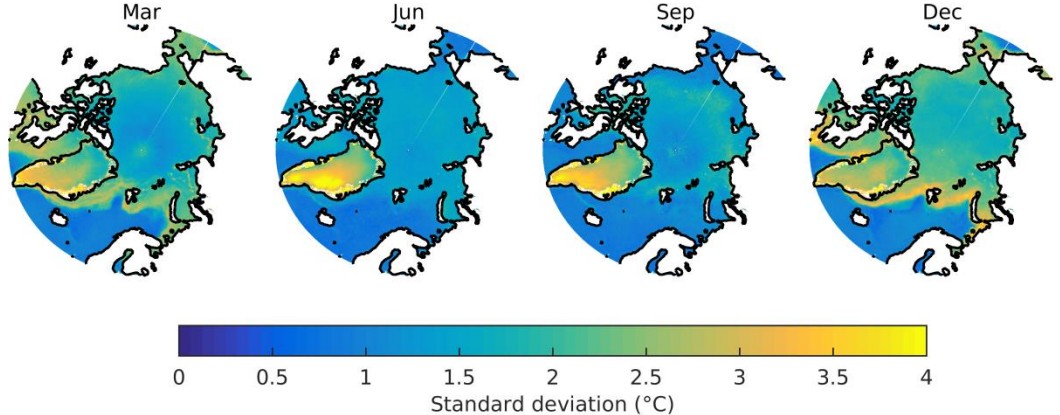

**Figure 4: Standard deviations of daily satellite surface temperature observations for March, June, September and December, averaged for the years 2000-2009 (°C).**

Additional satellite versus in situ differences arise when comparing satellite observations with pointwise ground measurements due to different spatial and temporal characteristics. To assess the magnitude of these effects, the $IST_{skin\_L3}$ data have been validated against in situ land ice temperatures from PROMICE and ARM stations. Table 2 shows the validation results of $IST_{skin\_L3}$ against in situ skin temperatures ($IST_{skin\_insitu}$) and in situ 2 meter air temperatures ($T2m_{insitu}$), respectively. The spatial and temporal sampling effects contribute to the overall uncertainty, but effects from erroneous cloud screening, algorithm simplifications, and uncertainties in the in situ observations are also included in the results. In general, $IST_{skin\_L3}$ correlates better with $T2m_{insitu}$ than with the $IST_{skin\_insitu}$. Moreover, the $IST_{skin\_L3}$-$T2m_{InSitu}$ difference shows smaller standard deviations than $IST_{skin\_L3}$-$IST_{skin\_insitu}$. However, as expected the biases and root mean squared differences (RMS) are larger for the $IST_{skin\_L3}$-$T2m_{insitu}$ differences than for the $IST_{skin\_L3}$-$IST_{skin\_insitu}$ differences. The reason is that the radiometric surface skin temperature can be significant different from the surface air temperature measurements (Adolph et al., 2018; Hall et al., 2008; Hudson and Brandt, 2005; Nielsen-Englyst et al., 2019; Vihma et al., 2008). On average, the skin temperature is colder than the air temperature, with the largest differences during clear-sky conditions and when the skin temperature is constrained by the melting point (melting snow has a maximum temperature of 0°C) (Nielsen-Englyst et al., 2019).

**Table 2. Validation of daily AASTI v.1 Level 3 IST ($IST_{skin\_L3}$) against in situ $IST_{skin}$ ($IST_{skin\_insitu}$) and T2m observations ($T2m_{insitu}$). N: number of matchups, Corr: correlation, Std: standard deviation, and RMS: root mean square difference.**

| | | $IST_{skin\_L3}$ - $IST_{skin\_insitu}$ | | | | $IST_{skin\_L3}$ − $T2m_{insitu}$ | | | |
| --- | --- | --- | --- | --- | --- | --- | --- | --- | --- |
| **Station** | **N** | **Corr** | **Bias** | **Std** | **RMS** | **corr** | **Bias** | **Std** | **RMS** |
| ARM_Atq | 1235 | 93.8 | -2.47 | 3.69 | 4.44 | 93.7 | -3.17 | 3.69 | 4.87 |





| | | | | | | | | | |
|---|---|---|---|---|---|---|---|---|---|
| ARM_Bar | 1594 | 94.1 | -0.73 | 4.30 | 4.36 | 94.6 | -1.14 | 4.02 | 3.86 |
| PROMICE KAN-M | 422 | 93.9 | -3.65 | 3.37 | 4.96 | 94.6 | -4.56 | 3.14 | 5.53 |
| PROMICE KAN-U | 239 | 93.9 | -1.75 | 3.32 | 3.75 | 94.4 | -3.39 | 3.17 | 4.64 |
| PROMICE KPC-U | 488 | 97.6 | -1.31 | 2.62 | 2.92 | 98.2 | -3.20 | 2.27 | 3.92 |
| PROMICE NUK-U | 296 | 77.7 | -4.09 | 5.00 | 6.45 | 84.7 | -7.19 | 4.01 | 8.23 |
| PROMICE QAS-U | 407 | 83.9 | -1.65 | 4.20 | 4.51 | 86.3 | -3.70 | 3.75 | 5.27 |
| PROMICE SCO-U | 403 | 91.5 | -4.60 | 4.25 | 6.26 | 93.7 | -7.55 | 3.75 | 8.43 |
| PROMICE TAS-U | 386 | 67.5 | -1.03 | 5.43 | 5.52 | 79.5 | -3.61 | 4.39 | 5.68 |
| PROMICE UPE-U | 125 | 88.2 | -3.13 | 3.88 | 4.97 | 90.0 | -5.49 | 3.50 | 6.50 |
| All data | 5595 | 92.9 | -2.03 | 4.24 | 4.70 | 93.2 | -3.36 | 4.12 | 5.32 |

# 3 Methods

## 3.1 Regression model

Nielsen-Englyst et al. (2019) analysed a large number of in situ stations with simultaneous T2m and $IST_{skin}$ observations and showed that empirical relationships existed between T2m and $IST_{skin}$. It was also shown, however, that the relationships

varied for the different regions. Based upon these results, it was decided to use a simple regression based method in this paper to derive the daily mean T2m from the satellite $IST_{skin\_L3}$ observations. Separate regression models have been derived for land ice and sea ice.

To test different types of regression models, the $IST_{skin\_L3}$ data have been matched up with in situ observations for each day (Høyer et al., 2018). This is done by requiring a distance to nearest in situ site of less than 0.5 degree (approximate 55 km in

latitude). All in situ observations, described in Sect. 2.1., have been matched up with $IST_{skin\_L3}$ data, resulting in a total number of daily matchups of 65,810 from 275 different observation sites (see Table 1). These have been divided into two subsets: one for training and one for validation of the different regression models for land- and sea ice, respectively. This has been done while ensuring similar coverage of training and validation data over the two domains, which is shown in Fig. 5. The result is that 13,792 observations are used for testing the regression models (and generating the regression coefficients)

and the remaining 20,872 observations are left for validation of the regression models over land ice. Over sea ice 15,035 observations are used for testing and 16,111 are left for validation.

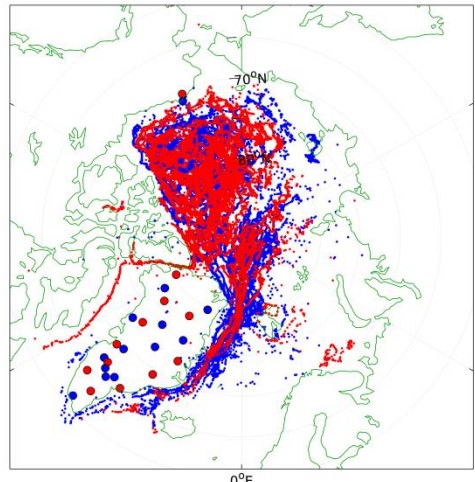

**Figure 5: Positions of matchups on sea ice and land ice (red: training, blue: validation)**

The regression model is based on multiple linear regression analysis using least squares (Menke, 1989). The multiple linear regression analysis equations can be written in matrix form,

$$d^{obs} = Gm + e \tag{4}$$

$$d^{pre} = Gm, \tag{5}$$

where $d^{obs}$ and $d^{pre}$ are vectors containing the observed and modelled in situ air temperatures, respectively, $G$ is a matrix containing the various predictors, $m$ is a vector containing regression coefficients, and $e$ is the fitting error.

The regression coefficients are found using damped least squares (Menke, 1989). The least squares method is used since the problem is generally over-determined, and the damping is added to limit effects of noisy data. The regression coefficients are thus given as:

$$G^{-g} = (G^T G + \varepsilon^2 I)^{-1} G^T \tag{6}$$

$$m = G^{-g} d^{obs}, \tag{7}$$

where $G^{-g}$ is called the generalized inverse, $\varepsilon$ is a damping factor and $I$ is an identity matrix (with ones in the diagonal and zeros elsewhere). The superscript operator T denotes transposing and -1 denotes inversion. We have tested a range of damping factors to assess the relation to the error coefficients. A damping factor of 0.2 was chosen to avoid overfitting noise in the data, while keeping the error coefficients low.

The choice of predictors is based on current knowledge of the parameters that influence the relationship between $IST_{skin}$ and $T2m_{insitu}$ (Nielsen-Englyst et al., 2019), limited by the available satellite data. Nielsen-Englyst et al. (2019) showed that on average $T2m_{insitu}$ is 0.65-2.65°C warmer than $IST_{skin}$ with variations depending on region (lower ablation zone, upper-middle ablation zone, accumulation zone, seasonal snow cover and sea ice). The T2m-Tskin difference varies over the day and season with smallest differences around noon and early afternoon during spring, fall and summer in non-melting conditions.





Nielsen-Englyst et al. (2019) also found that at the observation sites located on the Arctic sea ice and snow covered regions of North Alaska the $T2m_{insitu}$-$IST_{skin}$ difference decreases almost linearly as a function of wind speed due to increased turbulent mixing of the air for higher wind speeds. Contrary, the maximum $T2m_{insitu}$-$IST_{skin}$ differences over the GrIS occur at wind speeds of about 5 m s$^{-1}$. This is also seen by Adolph et al. (2018) at Summit, GrIS and by Hudson and Brandt (2005)

at the South Pole, and the feature is related to the pronounced katabatic winds in these regions. Furthermore, Nielsen-Englyst et al. (2019) found that the $T2m_{insitu}$-$IST_{skin}$ difference tends to decrease linearly as a function of the cloud cover fraction for all seasons and all regions. The reason for this is that clouds have a predominately warming effect on the skin temperature in the Arctic (Intrieri, 2002; Walsh and Chapman, 1998). Nielsen-Englyst et al. (2019) showed an almost linear relationship between the $T2m_{insitu}$-$IST_{skin}$ difference and the $IST_{skin}$, with larger differences for colder skin temperatures. Based on these

findings we have calculated the correlations between satellite skin temperature ($IST_{skin\_L3}$), in situ surface air temperatures ($T2m_{insitu}$), latitude (Lat), downward shortwave radiation (SWd) and not considering clouds (theoretical), and wind speed (WS) from ERA-Interim reanalysis. Since the cloud cover fraction and longwave radiation are unknown in this case, we have tested $IST_{skin\_L3}$ as a predictor instead. The resulting correlations are shown in Table 3.

**Table 3 Correlations between satellite-measured $IST_{skin\_L3}$, in situ measured $T2m_{insitu}$, latitude (Lat), theoretical downward**
**shortwave radiation (SWd), and ERA-Interim wind speed (WS).**

|  |  | $IST_{skin\_L3}$ | $T2m_{insitu}$ | Lat | SWd | WS |
|---|---|---|---|---|---|---|
|  | $IST_{skin\_L3}$ | 1.00 | 0.96 | -0.22 | 0.72 | -0.25 |
|  | $T2m_{insitu}$ | 0.96 | 1.00 | -0.25 | 0.61 | -0.28 |
| Land ice | Lat | -0.22 | -0.25 | 1.00 | -0.05 | 0.10 |
|  | SWd | 0.72 | 0.61 | -0.05 | 1.00 | -0.23 |
|  | WS | -0.25 | -0.28 | 0.10 | -0.23 | 1.00 |
|  | $IST_{skin\_L3}$ | 1.00 | 0.96 | -0.07 | 0.79 | -0.06 |
|  | $T2m_{insitu}$ | 0.96 | 1.00 | -0.03 | 0.74 | -0.07 |
| Sea ice | Lat | -0.07 | -0.03 | 1.00 | 0.03 | -0.04 |
|  | SWd | 0.79 | 0.75 | 0.03 | 1.00 | -0.21 |
|  | WS | -0.06 | -0.07 | -0.04 | -0.21 | 1.00 |

The $IST_{skin\_L3}$ and air temperatures are well correlated (above 90% correlation), and $IST_{skin\_L3}$ also show correlation with the shortwave radiation. Part of the correlation between temperature and the theoretical shortwave radiation is expected to be due to correlation of a seasonal cycle in both signals, not necessarily indicating causality. Therefore, for the regression

modelling, a seasonal cycle with fit of amplitude and phase was also tested. A total of 5 regression models with different predictors have been tested (Høyer et al., 2018):





$\hat{\text{IST}}_{\text{skin}}$: 
$$T2m_{sat} = \alpha_0 + \alpha_1 IST_{skin\_L3} \qquad (8)$$

$\hat{\text{IST}}_{\text{skin}}$SWd: 
$$T2m_{sat} = \alpha_0 + \alpha_1 IST_{skin\_L3} + \alpha_2 SWd \qquad (9)$$

$\hat{\text{IST}}_{\text{skin}}$WS: 
$$T2m_{sat} = \alpha_0 + \alpha_1 IST_{skin\_L3} + \alpha_2 WS \qquad (10)$$

$\hat{\text{IST}}_{\text{skin}}$Lat: 
$$T2m_{sat} = \alpha_0 + \alpha_1 IST_{skin\_L3} + \alpha_2 Lat \qquad (11)$$

$\hat{\text{IST}}_{\text{skin}}$Season: 
$$T2m_{sat} = \alpha_0 + \alpha_1 IST_{skin\_L3} + \alpha_2 \text{COS}\left(\frac{t \cdot 2\pi}{1\,yr}\right) + \alpha_3 \text{SIN}\left(\frac{t \cdot 2\pi}{1\,yr}\right) \qquad (12)$$

The regression model in Eq. (8) is limited to an offset and a scaling of $IST_{skin\_L3}$, while all other regression models also have a third predictor. The model in Eq. (9) uses theoretical shortwave radiation, Eq. (10) uses the wind forcing, Eq. (11) uses latitude variation, and Eq. (12) uses a seasonal variation. In the regression model in Eq. (12), the seasonal variation is

assumed to be the shape of a cosine function, $A \cdot \text{COS}\left(\frac{t \cdot 2\pi}{1\,yr} - \varphi\right)$, where A is the amplitude, $\varphi$ is the phase and t is time. Since $\text{COS}(x_1 - x_2) = \text{COS}(x_1)\text{COS}(x_2) + \text{SIN}(x_1)\text{SIN}(x_2)$, the seasonal cycle can be rewritten to the form in Eq. (12) with $A = \sqrt{\alpha_2{}^2 + \alpha_3{}^2}$ and $\varphi = \text{ARCTAN}\left(\frac{\alpha_3}{\alpha_2}\right)$.

The training data have been used to calculate the regression coefficients for each regression model covering land ice and sea ice, respectively. The training data have been used to investigate the performance of each regression model and the results

are shown in Table 4. The best correlation is found by using the regression model where T2m$_{sat}$ is predicted from IST$_{skin\_L3}$ combined with a seasonal variation ($\hat{\text{IST}}_{\text{skin}}$Season). This model predicts T2m$_{sat}$ better compared to the other regression models for both surface types, with correlations above 96 % and RMS values of 3.25-3.28°C against training data for both surface types (Table 4). In the following we will use the regression model given in Eq. 12 with the seasonal term included and with separate regression coefficients for land ice and sea ice, respectively. The values are shown in Table 5. The phase

corresponds to a maximum the 19[th] January and 12[th] February for land ice and sea ice, respectively. This is in agreement with Nielsen-Englyst et al. (2019) who found the strongest clear-sky inversion during the winter months (Dec-Feb) for all sites included in the analysis except from the ones located in the lower ablation zone (not included here), where pronounced surface melt takes place for long periods of time.

**Table 4: Statistics on the relation between observed and modelled temperatures for the training data. N: number of matchups used for testing, Corr: correlation, RMS: root mean square difference. Since the training data are used for the regression, the bias is 0 and thus the standard deviation equals RMS.**

| | N | Corr (%) | RMS (°C) |
|---|---|---|---|
| $\hat{\text{IST}}_{\text{skin}}$ | 13792 | 95.7 | 3.51 |
| $\hat{\text{IST}}_{\text{skin}}$SWd | 13792 | 96.2 | 3.28 |





| | | | | |
|---|---|---|---|---|
| Land ice | $\hat{I}ST_{skin}WS$ | 13792 | 95.8 | 3.47 |
| | $\hat{I}ST_{skin}Lat$ | 13792 | 95.8 | 3.48 |
| | $\hat{I}ST_{skin}Season$ | 13792 | 96.3 | 3.28 |
| | $\hat{I}ST_{skin}$ | 15035 | 96.0 | 3.32 |
| | $\hat{I}ST_{skin}SWd$ | 15035 | 96.0 | 3.32 |
| Sea ice | $\hat{I}ST_{skin}WS$ | 15035 | 96.0 | 3.32 |
| | $\hat{I}ST_{skin}Lat$ | 15035 | 96.1 | 3.28 |
| | $\hat{I}ST_{skin}Season$ | 15035 | 96.2 | 3.25 |

**Table 5: Model regression coefficients for $\ddot{I}ST_{skin}Season$.**

| | Offset, $\alpha_0$ (°C) | $IST_{skin\_L3}$ factor, $\alpha_1$ | Amplitude, A | Phase, $\varphi$ |
|---|---|---|---|---|
| Land ice | 4.20 | 1.06 | 2.26 | -0.33 |
| Sea ice | 1.46 | 0.89 | 1.83 | -0.75 |

## 3.2 Uncertainty estimates for T2m$_{sat}$

Uncertainty estimates on the derived T2m$_{sat}$ are crucial to facilitate the usage of the data set in modelling and for monitoring purposes. The uncertainty estimates of the satellite-derived T2m$_{sat}$ data follow the approach in Bulgin et al. (2016) and Rayner et al. (2015), which has also been used for the AASTI data. The uncertainty on a single T2m$_{sat}$ estimate is divided into random, locally correlated and systematic uncertainty components, with the total uncertainty $\mu_{total\_t2m}$ given as the square root of the sum of the three squared components:

$$\mu_{total\_T2m} = \sqrt{\mu_{rnd\_T2m}{}^2 + \mu_{local\_T2m}{}^2 + \mu_{glob\_T2m}{}^2}$$

The random uncertainty component for the T2m$_{sat}$ belonging to a particular grid cell at a particular point in time is found by propagating the AASTI IST$_{skin\_L3}$ random uncertainty through the regression model:

$$\mu_{rnd\_T2m} = \sqrt{\left(\alpha_1 \mu_{rnd\_L3}\right)^2},$$

with $\mu_{rnd\_L3}$ given as the aggregated $\mu_{rnd\_L2}$:

$$\mu_{rnd\_L3} = \frac{\mu_{rnd\_L2}}{\sqrt{N}},$$

where $N$ is the number of observations for each bin in the aggregation from L2 to L3. The $\sqrt{N}$ reduction applies because the random uncertainty of each L2 data point that goes into the L3 calculation is by definition independent from the other.





The L3 global uncertainty component does not average out in any aggregation and is thus transferred directly from the L2 uncertainty estimate and has been multiplied by $\alpha_1$ to make up $\mu_{glob\_T2m}$:

$$\mu_{glob\_T2m} = \alpha_1 \mu_{glob\_L3} = \alpha_1 \cdot 0.1°C$$

The $\mu_{local\_T2m}$ contains both the local uncertainty component of L2, a sampling error $\mu_{lsamp\_L3}$ related to sampling errors in space and time due to the aggregation, a relationship error, cloud mask uncertainty etc. When aggregating from L2 to daily L3, additional sources of uncertainty enter through the gridding process as $IST_{skin\_L3}$ can only be retrieved for clear-sky pixels. This introduces a temporal and spatial sampling uncertainty. If all our satellite observations were obtained during all-sky conditions we assume that the high polar temporal coverage is such that the temporal sampling uncertainty in the L3 files can be set to zero. However, this is not the case and using only clear-sky observations generally leads to a clear-sky bias in averaged $IST_{skin}$ satellite observations when compared to in situ observations (Hall et al., 2012; Nielsen-Englyst et al., 2019; Rasmussen et al., 2018). The relationship error represents the standard deviation of the residuals calculated at in situ stations, where both skin and air temperatures are available, i.e. $T2m_{sat}$-$T2m_{insitu}$. Estimating all the different components that make up the $\mu_{local\_T2m}$ is a very challenging task and is out of the scope of this paper. Instead, we estimate the $\mu_{local\_T2m}$ component using a simple regression model fitted to the satellite derived T2m and in situ T2m differences. Separate models have been chosen for the land ice and sea ice, due to the differences in the error characteristics. The variables to include in the uncertainty regression models have been chosen from a careful examination of the matchup data set. For land ice and sea ice the most relevant variables were the $IST_{skin\_L3}$ itself and the number of 3 h time bins with observations in the L3, $N_{bins}$.

For land ice the regression model for $\mu_{local\_T2m}$ is given as following:

$$\mu_{local\_T2m\_landice} = \beta_0 + \beta_1 IST_{skin\_L3} + \beta_2 N_{bins},$$

while the regression model for sea ice is given as:

$$\mu_{local\_T2m\_seaice} = \gamma_0 + \gamma_1 IST_{skin_{L3}} + \gamma_2 IST_{skin_{L3}}^2 + \gamma_3 N_{bins}.$$

The coefficients have been determined by fitting to the $T2m_{sat}$-$T2m_{insitu}$ standard deviations calculated for the training data with $IST_{skin\_L3}$ bin intervals of 2°C and $N_{bins}$ interval of 1. The $\mu_{rnd\_T2m}$ and $\mu_{glob\_T2m}$ components have been removed from the standard deviations in each bin as well as an assumed in situ uncertainty of 0.1°C and an average sampling uncertainty of 0.5°C (Høyer et al., 2017a; Reeves Eyre and Zeng, 2017) before fitting the regression models. The optimal regression coefficients for each domain are listed in Table 6.

**Table 6: Uncertainty model regression coefficients**

| Land ice | $\beta_0 = 3.82°C$ | $\beta_1 = -0.24$ | $\beta_2 = -0.03$ | |
|---|---|---|---|---|
| Sea ice | $\gamma_0 = 2.01°C$ | $\gamma_1 = -0.06$ | $\gamma_2 = -0.12$ | $\gamma_3 = -0.001$ |



# 4 Results

In Sect. 3.1 we selected the best (Eq. 12) of the 5 different algorithms and the derived coefficients (Table 4 and 5) to retrieve T2m from satellite surface temperature estimates. The dataset consists of daily estimates of mean air temperature on a 0.25 degree regular latitude-longitude grid, during the period 2000-2009 (Høyer et al., 2018; Kennedy et al., 2019). Each

temperature estimate is associated with three components of uncertainty: random uncertainties on the 0.25 degree daily scale, synoptic scale correlated uncertainty and globally correlated uncertainty excluding uncertainties related to the masking of clouds. The three types of uncertainties are also gathered in a total uncertainty estimate. The land ice temperatures have been calculated for grid cells categorized as ice shelf by ETOPO1, averaged to the 0.25 degree grid (Amante and Eakins, 2009). Sea ice temperatures have been calculated for grid cells with sea ice concentrations above 30 %, according to OSISAF

(Tonboe et al., 2016).

An evaluation of the product and the T2m$_{sat}$ regression model performance has been carried out by a comparison to the independent in situ data (i.e. validation subset described in Sect. 3.1). Figure 6 shows an example of daily near surface air temperatures on Jan 1[st], 2008. Circles are in situ T2m measurements from coincidence independent AWSs and buoys. The overall model performance when compared to all independent AWS and buoy observations is summarized in Table 7. The

satellite derived air temperatures are about 0.3°C warmer than measured in situ air temperature for both land ice and sea ice. The correlations are above 95 % for both surface types and the RMS is 3.47°C and 3.20°C for land and sea ice, respectively. Note that the uncertainty of the in situ data is also included in these RMS values.

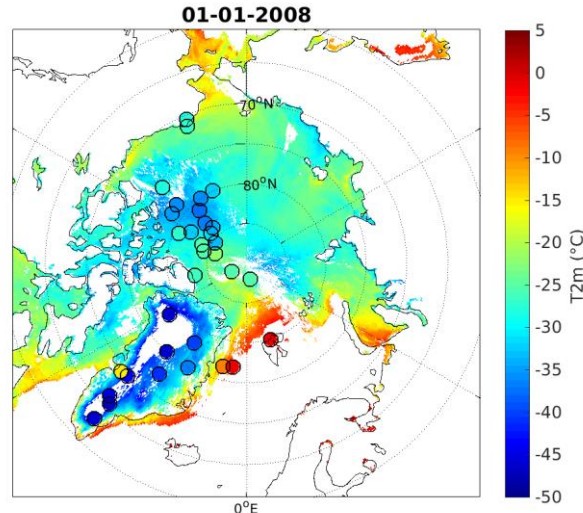

**Figure 6: Daily mean air surface temperature over land ice and sea ice from January 1, 2008. Circles show in situ measurements.**





**Table 7: Statistics on the relation between satellite-derived and in situ measured temperatures for comparison with independent validation data. N: number of matchups used for validation, Corr: correlation, bias: T2m$_{sat}$ − T2m$_{insitu}$ difference, Std: standard deviation, RMS: root mean square difference.**

|          | N     | Corr (%) | bias (°C) | Std (°C) | RMS (°C) |
|----------|-------|----------|-----------|----------|----------|
| Land ice | 20872 | 95.5     | 0.30      | 3.45     | 3.47     |
| Sea ice  | 16111 | 96.5     | 0.35      | 3.18     | 3.20     |

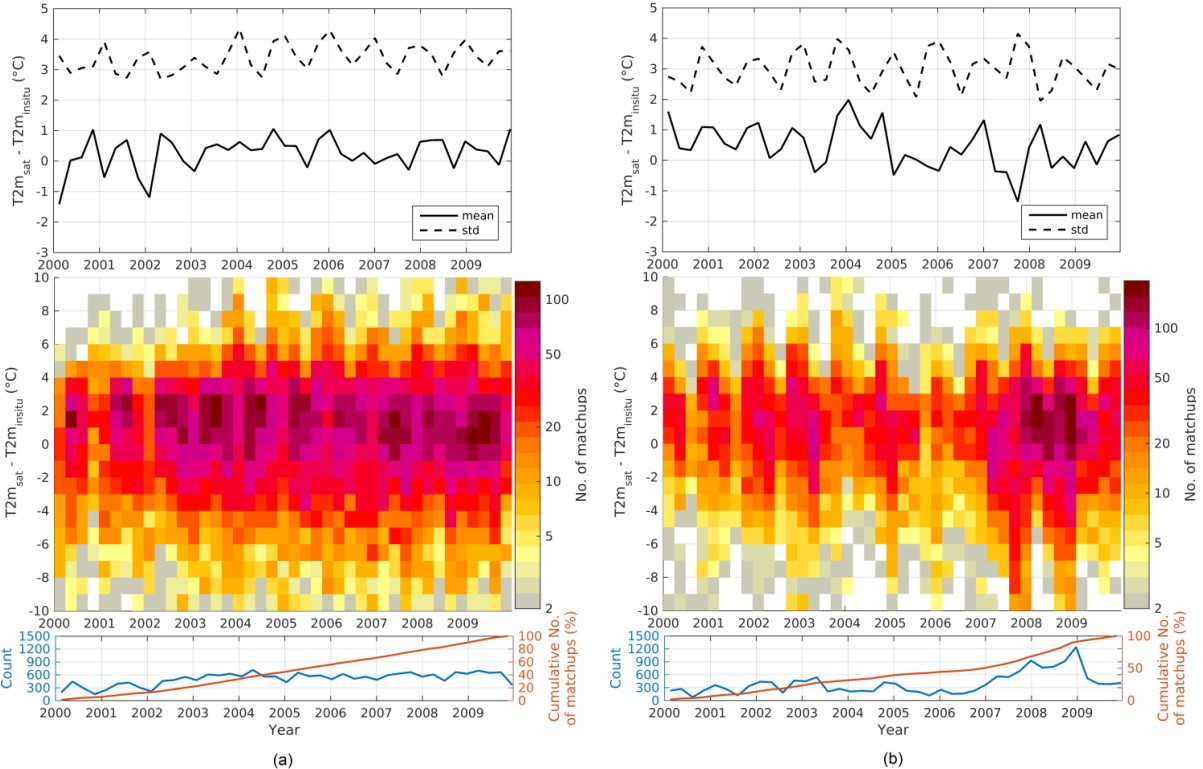

**Figure 7: Estimated T2m minus observed T2m (bin size of 1°C) for the full time period (bin size of 90 days) for (a) land ice and (b) sea ice, respectively. The dashed lines are standard deviations while the solid lines are bias in the upper figure. The surface plots in the middle figures show the number of matchups in each bin, while the bottom plots show the number of matchups (blue) and the cumulative percentage of matchups (red) in each time bin.**

Figure 7 shows the seasonal averaged independent validation statistics for land ice and sea ice, respectively. For both land ice and sea ice there are a seasonal dependency in standard deviation with largest values during winter and smallest during summer. This is probably explained by a better cloud screening performance during sunlit periods (Karlsson and Dybbroe, 2010). No significant seasonal cycle is seen in the mean bias, except for the sea ice region during 2000-2004, where there is a tendency to a warm bias during December and January.



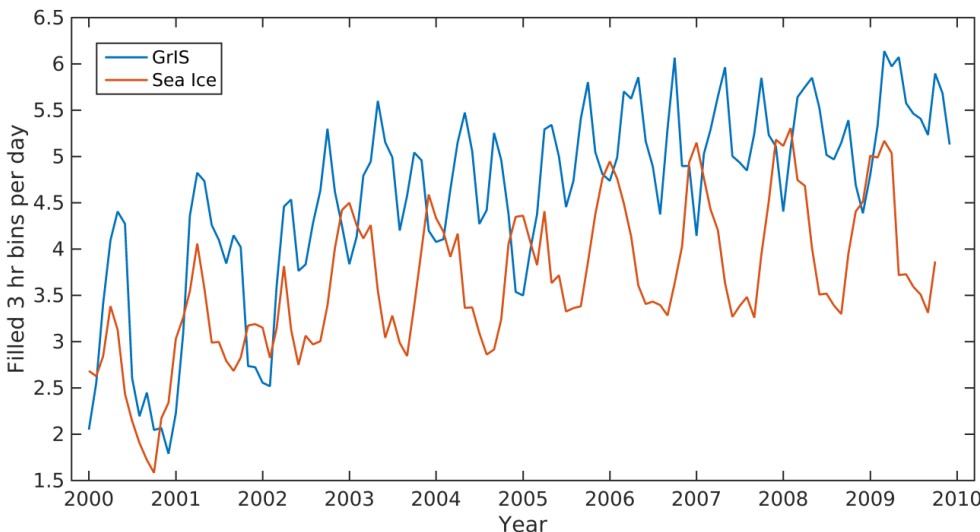

**Figure 8: Average number of filled 3 h bins per day for the Greenland Ice Sheet and the Arctic Sea Ice, respectively.**

As more satellite observations have become available over the time period a better coverage of the surface temperature is expected over time. Figure 8 shows the average number of filled 3 h bins per day for the GrIS and Arctic Sea Ice, 2000-

2009. Both surface types show an increase in filled 3 h bins over time, with large seasonal variations. In most years sea ice has 1-1.5 filled bins per day more during winter than summer, due to a more extensive cloud cover over sea ice during summer. The GrIS typically has fewer filled bins per day during winter and summer, than spring and fall, which is also explained by differences in cloud coverage.

Figure 9 shows T2m$_{sat}$-T2m$_{insitu}$ differences plotted as a function of AASTI L3 skin temperature for land ice and sea ice,

respectively. Over land ice the standard deviation decreases as a function of IST$_{skin\_L3}$, while the bias is around zero for IST$_{skin\_L3}$ between -45°C and -10°C, and positive for higher temperatures and negative for lower temperatures. For sea ice the maximum standard deviation is found at skin temperatures of about -20°C, with smaller standard deviations for higher and lower IST$_{skin\_L3}$. Positive biases are found for very cold skin temperatures (< -25°C) and for temperatures around the melting point (> -4°C), while the intermediate temperatures have a slightly negative bias. This effect is included in the uncertainty

estimates as presented in Sect. 3.2, which include IST$_{skin\_L3}$ as a predictor for both land ice and sea ice.

Figure 10 shows the validation results of the estimated uncertainties, where the T2m$_{sat}$-T2m$_{insitu}$ difference is plotted against the theoretical total uncertainties as obtained in Sect. 3.2 for land ice and sea ice, respectively. The dashed lines represent the ideal uncertainty with the assumptions that the in situ observations have an uncertainty of 0.1°C and that the sampling uncertainty is 0.5°C. The estimated uncertainties show good agreement with the observed uncertainties for both land ice and

sea ice, when the error bars follow the dashed line, which is the case here.

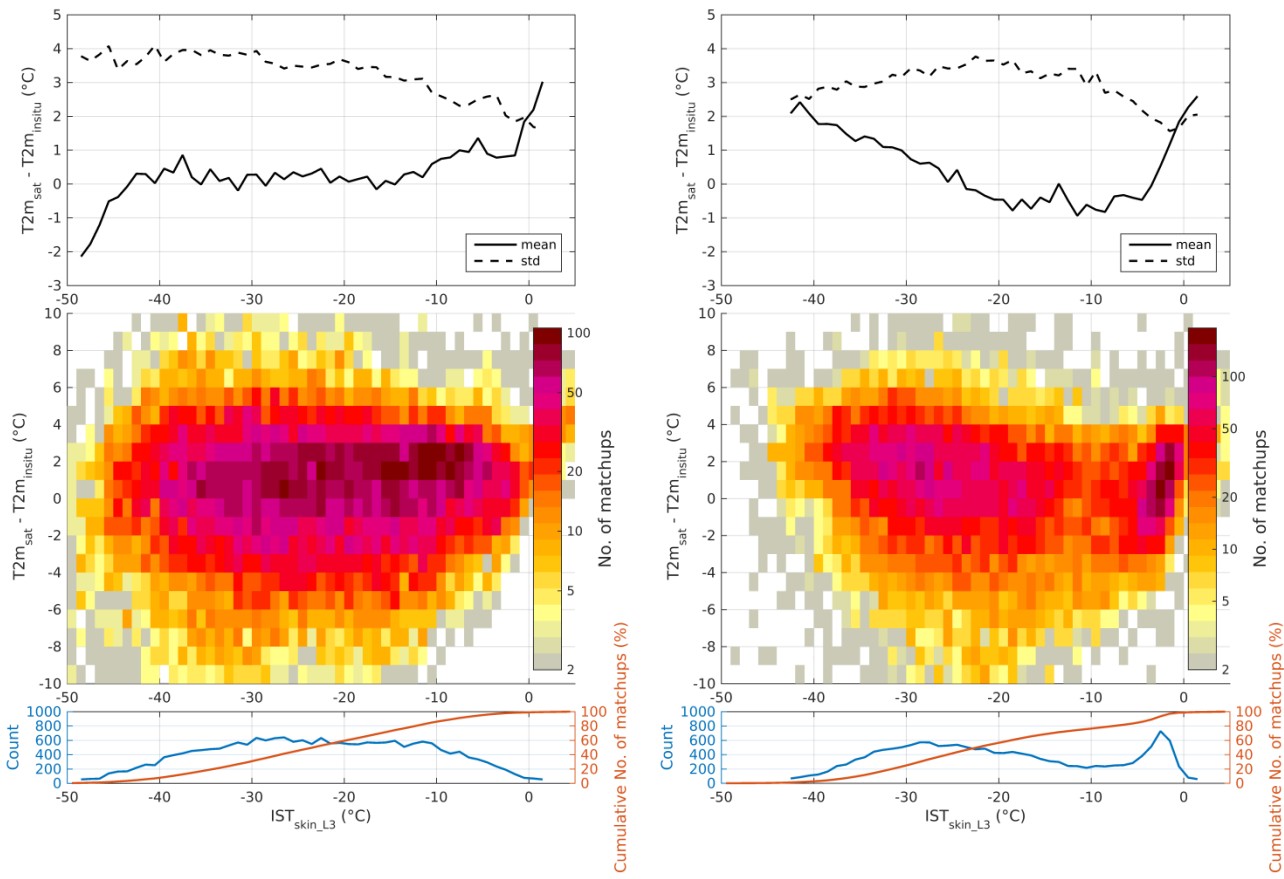

**Figure 9: Estimated T2m minus observed T2m (bin size of 1°C) as a function of binned (bin size of 1°C) satellite IST$_{skin\_L3}$ for (a) land ice and (b) sea ice, respectively. The dashed lines are standard deviations while the solid lines are bias in the upper figure. The surface plots in the middle figures show the number of matchups in each bin while the bottom plots show the number of matchups (blue) and the cumulative percentage of matchups (red) in each IST$_{skin\_L3}$ bin.**



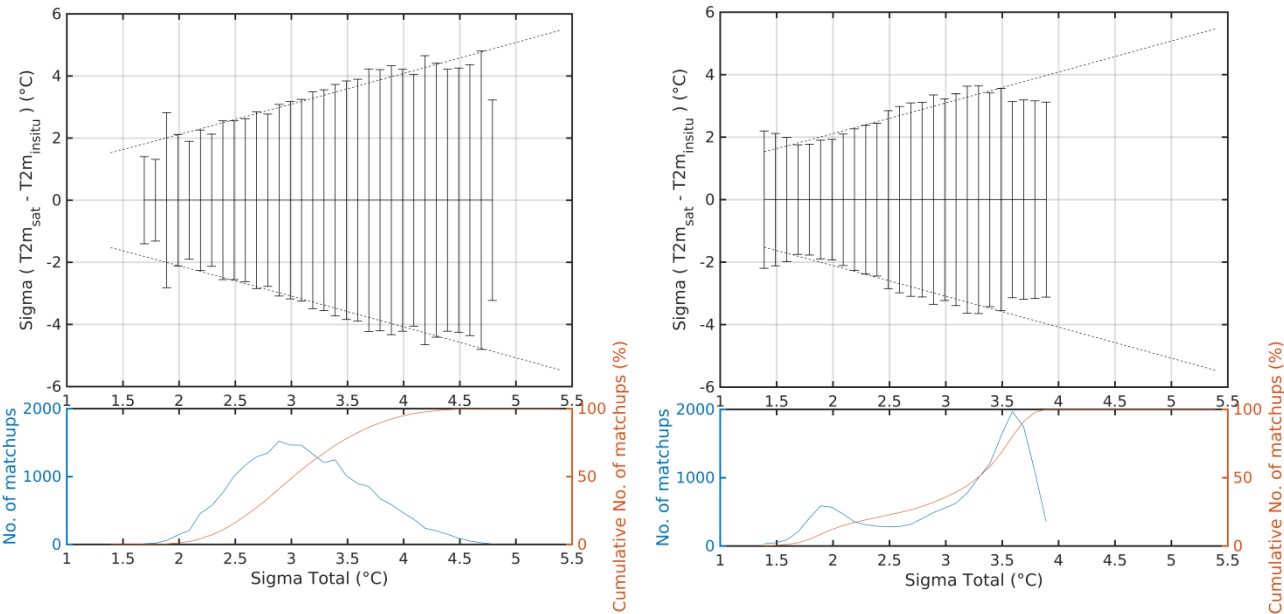

**Figure 10: Satellite estimated T2m uncertainty validation with respect to independent in situ T2m for (a) land ice and (b) sea ice. Dashed lines show the modelled uncertainty accounting for uncertainties in the in situ T2m and the sampling error. Solid black lines show one standard deviation of the estimated minus in situ differences for each 0.1 °C bin. The bottom plots show the number of matchups (blue) and the cumulative percentage of matchups for each bin (red).**

The performance of T2m$_{sat}$ has been compared to the performance of T2m from ECMWF's reanalysis ERA-Interim (T2m$_{ERA}$; Dee et al., 2011). Table 8 shows the performance of T2m$_{ERA}$ against the independent in situ T2m observations, which should be compared with the performance of the regression derived T2m$_{sat}$ as shown in Table 7. The comparison may not be truly independent as a number of stations and buoys have been assimilated into the ERA-Interim data product (Dee et al., 2011), which would favour the ERA Interim in the comparison. Yet, the bias is significantly lower for T2m$_{sat}$ than for T2m$_{ERA}$, and the other validation parameters are similar, with slightly better correlation and standard deviation, but slightly worse RMS results for T2m$_{ERA}$.

**Table 8: Statistics on the relation between ERA-Interim and in situ measured temperatures for independent test data. N: number of matchups used for validation, Corr: correlation, bias: T2m$_{sat}$ − T2m$_{insitu}$ difference, Std: standard deviation, RMS: root mean square difference.**

|  | N | Corr (%) | Bias (°C) | Std (°C) | RMS (°C) |
|---|---|---|---|---|---|
| Land ice | 20872 | 96.4 | 3.41 | 3.18 | 4.66 |
| Sea ice | 16111 | 96.9 | 1.14 | 3.02 | 3.22 |





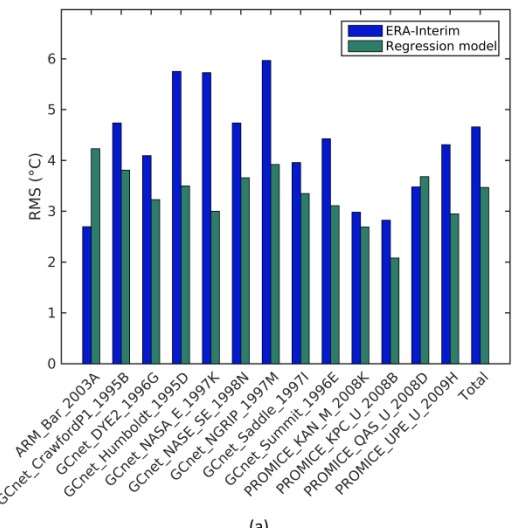
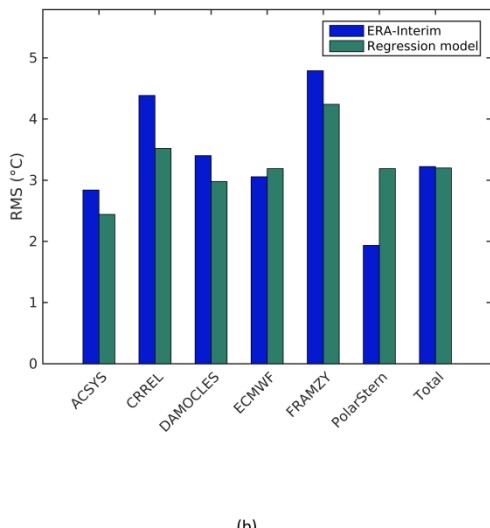

(a)                                                (b)

**Figure 11: Root mean square (RMS) differences calculated for (a) land ice sites and (b) sea ice sites. Blue bars are RMS obtained by comparing in situ with ERA-Interim T2m, while green bars are in situ compared with the regression derived T2m. Only buoys with more than 200 observations are included. The last two bars listed as "total" are the RMS obtained by using all validation data.**

Figure 11 gives an indication of the performance of $T2m_{sat}$ and $T2m_{ERA}$ at individual sites for each surface type. It shows the RMS difference between in situ measured T2m and (a) $T2m_{sat}$ and (b) $T2m_{ERA}$ for the independent test sites and for both surface types. Due to the large number of buoys these have been validated for each data source with all observations weighted equally. The last bars refer to the RMS obtained by validating all test sites in one long time series weighting all

10  daily observations equally. The total $T2m_{sat}$ agrees better with in situ observations for both surface types compared to ERA-Interim. Over the GrIS $T2m_{sat}$ performs better than $T2m_{ERA}$, while ERA-Interim agrees better with in situ observations from the North Alaska site, Barrows. Over sea ice $T2m_{ERA}$ agrees better with in situ observations from ECMWF data stream and Polarstern. However, these may be assimilated into ERA-Interim. The independent in situ observations by ACSYS, CRREL, DAMOCLES and FRAMZY are better reproduced by the satellite-derived T2m.

15  The monthly mean near surface air temperature estimates averaged over the GrIS have been shown in Fig. 12 for 2000-2009. The GrIS records a distinct annual cycle in near surface air temperature. The monthly mean air temperature typically reaches a maximum of -4°C during July and a minimum of about -28°C during winter. As is common for the Arctic environment, the temporal variability is largest during winter due to a more vigorous atmospheric circulation (Steffen, 1995).


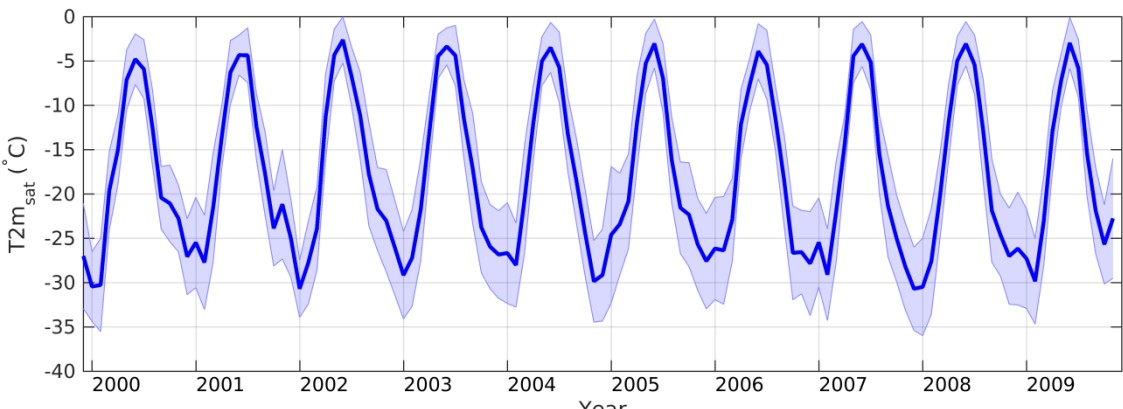

**Figure 12: Monthly mean T2m$_{sat}$ for the Greenland Ice Sheet. The shading represents the variability.**

The monthly mean T2m$_{sat}$ is shown in Fig. 13 for March, June, September and December averaged over the period 2000-

2009. The interior and northern part of the GrIS is typically colder than other parts of the Arctic in all months, while the

warmest regions are found along the sea ice marginal ice zone and the ablation zone of the GrIS. During summer little spatial

variability in monthly mean T2m is found over the Arctic sea ice.

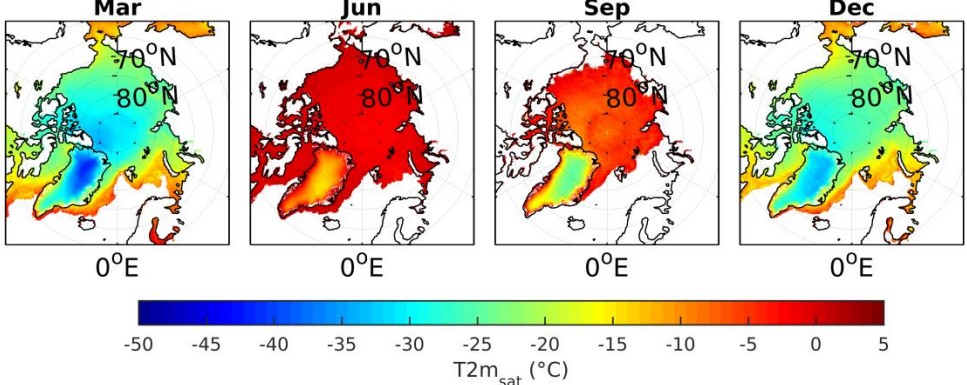

**Figure 13: Monthly mean T2m$_{sat}$ during March, June, September and December, averaged for the period 2000-2009.**

**5 Discussion**

Due to the limited number of in situ observations in the Arctic, and especially over sea ice, it is not a simple task to gather in

situ observations for testing and validating the regression models. The sparse number of in situ observations over sea ice is

the greatest challenge, as also discussed in Nielsen-Englyst et al., 2019. The lack of observations that represent all conditions

and regions in the Arctic limits the details to which the T2m versus IST$_{skin}$ relationship model can be derived and validated.

As infrared satellites cannot measure the surface temperature during cloudy conditions a cold clear-sky bias is often

observed in infrared satellite IST$_{skin\_L3}$ averages compared to all-sky temperature averages. When using satellite IST$_{skin\_L3}$





observations it is thus important to assess the clear-sky bias, which varies with different temporal averaging windows (Nielsen-Englyst et al., 2019). However, through the use of an empirical statistical method, which is trained against daily averaged in situ 2 m air temperatures, the conversion from $IST_{skin\_L3}$ to $T2m_{sat}$ removes the systematic $IST_{skin\_L3}$ clear-sky bias effects that may be present in the satellite data set. As a result, we obtain a $T2m_{sat}$ estimate which performs similar to the

$IST_{skin\_L3}$ when compared against in situ observations.

For short-lasting (<24 hours) cloudy conditions the division into 3 h bin averages and the requirement of filled 3 h bins both during night (18-6 local solar time) and day (6-18 local solar time) ensure that the diurnal cycle is best resolved despite the gaps with clouds. For long-lasting (>= 24 hours) cloudy conditions $IST_{skin\_L3}$ is not available and a statistical technique or the use of atmospheric models and assimilation may fill in the gaps.

Previous studies show a strong dependence of wind speed for both land ice and sea ice, but with different dependencies (Adolph et al., 2018; Hudson and Brandt, 2005; Miller et al., 2013; Nielsen-Englyst et al., 2019). However, the performance of the satellite derived T2m product did not improve when including the wind speed information from ERA Interim. The reason is that the quality of the ERA Interim wind speed is not adequate for use in the relationship model. Especially, the representation of katabatic winds in numerical weather prediction (NWP) models is a challenging task due to a high

resolution needed in the vertical (Grisogono et al., 2007; Steeneveld, 2014; Weng and Taylor, 2003; Zilitinkevich et al., 2006), but also the processes of snow surface coupling, radiation and turbulent mixing are hampered by limited resolution, while their relative importance varies with wind speed (Sterk et al., 2013). More accurate information on the wind speed would very likely improve the performance of the regression model when including wind speed as predictor. In particular, the higher resolution NWP output may be very beneficial in the regions of the GrIS, where the local topography interacts

with the wind through katabatic effects (DuVivier and Cassano, 2013; Oltmanns et al., 2015; Renfrew, 2004). At the time of the present work, the ERA5 analysis was not available (Copernicus Climate Change Service (C3S), 2017), but this may bring improvements in future work. Moreover, regional high resolution reanalysis are currently being carried out within the Copernicus Arctic regional Reanalysis service C3S project (https://climate.copernicus.eu/copernicus-arctic-regional-reanalysis-service). It is likely that such products will provide winds that can be used within a relationship model.

The $T2m_{SAT}$ data set developed here only covers the Arctic, but the AASTI data set also covers the Antarctica. This implies that similar statistical methods can be derived for the Antarctic ice cap and sea ice. Preliminary investigations indicate that a T2m product can be derived for the ice caps with similar performance as for the GrIS, whereas the Southern Ocean sea ice is challenging due to very few in situ observations (Morice et al., 2012). For both Southern regions, more in situ observations are needed to repeat the work performed for the Arctic and to determine a reliable statistical model.

Including other available satellite products, such as Modis IST observations (Hall et al., 2004) or the (A)ATSR data set (Ghent et al., 2017) could improve upon the quality of the $T2m_{SAT}$ product. However, adding new data requires a detailed knowledge of the characteristics of the data set, such as sampling frequency and uncertainty of the IST observations. In addition, determination of the relationship model is needed again. At the same time, adding more satellite overpasses to the daily estimates may not improve the uncertainty of the products. This is evident when comparing Fig. 7 and 8 where the



variation in the number of satellite observations during the record (Fig. 8) is not reflected in a similar variation in the performance of the product (Fig. 7). The uncertainty in the beginning of the record is comparable to the uncertainty in the end of the record, despite an almost doubling of the observed 3 hourly averages throughout the day.

The AASTI version builds upon the Clara version 1 data set from the CM-SAF. A version 2 of the data set is now available
(Karlsson et al., 2017), facilitating the production of an AASTI version 2 data set that covers from 1982 up to present. With a consistency in the retrieval algorithm and data sets, it will be possible to use the relationship model to produce a satellite based climate data record of T2m from 1982 to today.

## 6 Conclusions

The air temperature over land ice and sea ice is an obvious indicator for Arctic climate change and it can easily be compared
with climate change indicators from other regions. This study introduces a methodology for using satellite skin temperatures for estimating air temperatures, to compensate for the lack of in situ measurements, and as a supplement to reanalysis products. Daily near surface air temperatures (T2m) have been estimated based on daily satellite Level 3 (L3) observations of ice surface skin temperatures ($IST_{skin\_L3}$) in the Arctic, using the Arctic and Antarctic ice Surface Temperatures from thermal Infrared satellite sensors (AASTI) reanalysis. A regression based method has been used and tuned against in situ
observed T2m using $IST_{skin\_L3}$ observations covering both Arctic sea ice and land ice. As explaining factors, latitude, theoretical downward shortwave radiation (not considering clouds), seasonal cycle, and wind speed (ERA-Interim reanalysis) were tested. These factors were selected based on current knowledge from the literature (Adolph et al., 2018; Hall et al., 2008; Hudson and Brandt, 2005; Nielsen-Englyst et al., 2019; Vihma and Pirazzini, 2005), limited by the available data. The seasonal cycle was introduced based upon the results from an analysis of in situ observations, where a seasonal
cycle in the relationship between surface skin and near surface temperature was observed (Nielsen-Englyst et al., 2019). The best correlation against the training data was found using a model where $T2m_{sat}$ is predicted from daily satellite $IST_{skin\_L3}$ combined with a seasonal variation assumed to have the shape of an annual harmonic. This model has been used to derive T2m from the AASTI ice surface skin temperatures over the Arctic during the period 2000-2009 (Kennedy et al., 2019), where different regression coefficients have been used for land ice and sea ice.

The estimated $T2m_{sat}$ data record has been validated against independent in situ measured 2 m air temperatures. The validation results indicate average correlations of 95.5% and 96.5% and average root mean square errors of 3.47°C and 3.20°C for land ice and sea ice, respectively. An uncertainty model has been developed and all daily $T2m_{sat}$ estimates come with a total uncertainty divided into a random, locally systematic and large-scale systematic uncertainty component. The total uncertainty of the satellite derived $T2m_{sat}$ shows good validation results when validated against independent in situ
observations.

The satellite derived $T2m_{sat}$ product has been compared to ERA-Interim T2m estimates and has proven to validate similar or better compared with ERA-Interim estimates. The $T2m_{sat}$ product is independent of the quality of the NWP forecasts and



thus represents an important alternative to the model based T2m. The regression models presented here both work on satellite observations that are available from reprocessed records but opens up for a near real time estimation of T2m from satellites. The results obtained for the ice covered areas show that there is a large potential for using satellite observed surface temperatures for estimating near surface air temperatures. However, these estimates are not supposed to replace the

already existing air temperature measurements, but rather to supplement these e.g. in areas where no in situ observations are currently available.

## 7. Data availability

The PROMICE data can be accessed through http://www.promice.dk (last access: 16 November 2018). The ARM data are available at https://www.archive.arm.gov/discovery/\#v/results/s/s::co (last access: 21 December 2018). GC-Net data can be

found through doi:10.5067/6S7UHUH2K5RI (Kindig, 2010). Data from CRREL mass balance buoys are available from: http://imb-crrel-dartmouth.org (last access: 24 November 2016), while POLARSTERN data can be downloaded at https://dship.awi.de/Polarstern.html (last access: 24 November 2016. FRAMZY data are available from doi:10.1594/WDCC/UNI_HH_MI_FRAMZY2002 (Brümmer et al., 2012b), http://dx.doi.org/doi:10.1594/WDCC/UNI_HH_MI_FRAMZY2007 (Brümmer et al., 2011b), and

http://dx.doi.org/doi:10.1594/WDCC/UNI_HH_MI_FRAMZY2008 (Brümmer et al., 2011c), while ACSYS data are found here: DOI: 10.1594/WDCC/UNI_HH_MI_ACSYS2003. Damocles data can be found here: doi:10.1594/wdcc/uni_HH_MI_DAMOCLES2007 (Brümmer et al., 2011a). The traditional buoy and ship data obtained from ECMWF are distributed through the World Meteorological Organization's (WMO) Global Telecommunication System (GTS) and available for members at the ECMWF Meteorological Archival and Retrieval System (MARS). Finally, the

AASTI IST$_{skin\_L2}$ data are available from doi:10.5285/60b820fa10804fca9c3f1ddfa5ef42a1 (Høyer et al., 2019). The derived surface air temperatures from satellite surface skin temperatures over ice can be downloaded from doi: 10.5285/f883e197594f4fbaae6edebafb3fddb3 (Kennedy et al., 2019).

## 8 Author contribution

Pia Nielsen-Englyst, Kristine S. Madsen and Gorm Dybkjær compiled and quality checked the in situ data. Pia Nielsen-Englyst, Jacob L. Høyer and Kristine S. Madsen designed and developed the regression model and estimated uncertainties. Gorm Dybkjær, Jacob L. Høyer and Rasmus Tonboe developed the AASTI IST$_{skin\_L2}$ data. Pia Nielsen-Englyst prepared the manuscript with contributions from all authors.





## 9 Competing interests

The authors declare that they have no conflict of interest.

## 10 Acknowledgements

This study was carried out as a part of the European Union Surface Temperatures for All Corners of Earth (EUSTACE),
which is financed by the European Union's Horizon 2020 Programme for Research and Innovation, under Grant Agreement
no 640171. The aim of EUSTACE is to provide a spatially complete daily field of air temperatures since 1850 by combining
satellite and in situ observations. The author would also like to thank the data providers.

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
