# Peer review of "Deriving Arctic 2 m air temperatures over snow and ice from satellite surface temperature measurements"

_The Cryosphere, 2019_

## Referee Comment (RC1) · Anonymous Referee #1 · 23 Oct 2019

The authors present a remote sensed 2m temperature product for the Arctic ocean and the Greenland Ice sheet. For this aim, they use Arctic and Antarctic Ice Surface Temperatures from thermal Infrared satellite Sensors (AASTI) data set and apply a correction to convert surface temperatures into 2 m temperatures. The derived temperatures are compared with in situ observations and the data set performance is compared to the performance of T2m from ERA-Interim.

First of all, I apologize for the long time I needed to complete this review, which added to the long time that the editor needed to find reviewers. Sadly, I'm not convinced that

the authors properly resolve all the challenges that occur when remote sensed surface temperatures are converted into t2m temperatures.

My largest objection is that the authors fail to properly resolve the "cloud problem". Infrared satellites can only measure surface temperatures in cloud free conditions. However, cloudy conditions lead totally different weather than cloud-free conditions, especially in winter. Too little attention is paid to this problem in the methodology and results section. A publishable data set must resolve the cloud-data-gap problem and it should be explicitly shown that reasonable estimates can be provided for cloudy conditions, even if these conditions last for days or more. As the authors do not aim to present a clear-sky T2m product, which would be of limited value, this shortcoming in the methodology and discussion must be resolved prior acceptance can be considered.

Specific major and minor comments:

P3L13: which "data"? I presume satellite data, as AWSs do not move. This must be clearly worded.

P4L10: Snow on sea ice have a major effect on the measured temperature as snow is a very good insulator (e.g. Graham et al, 2019, https://doi.org/10.1038/s41598-019-45574-5). Hence, if the buoy thermistor has a smaller diurnal cycle as the T2m sensor, snow cover is affecting the observation and buoy thermistor should be discarded as valid surface temperature observation. Unless "unrealistic data artifacts" includes damped daily cycles - which then should be stated explicitly -, I believe data scarcity should not be an excuse for retaining incorrect data.

P5L4: All three citations listed to introduce the AASTI refer to technical documents; they do not refer to peer reviewed papers. This is not extremely relevant in itself, but it raises, in my humble opinion, the necessity that the authors restate briefly

the methods to compile this dataset. Furthermore, from the title of Høyer 2019, this dataset is only providing clear-sky ice surface temperatures. This must be restated when the AASTI dataset is introduced.

P6L10: It does not become clear to me how these 3-hourly bin averages are aggregated into one daily value. The procedure should be added and described plainly.

P7F3: Although the figure is somewhat instructive, I would be more interested to see (also) **a)** the ratio between cloudy and cloud free observations, as the current figure is clouded by the variations in observation density and **b)** the percentage of days with one or more valid observations within every time interval. Increasing from 1 to 25 observations in a 3-hour interval improves the measurement accuracy, decreasing from 1 to 0 leads to a data gap.
By the way, I am puzzled by the fact that even far North (>75 N), where the polar night and midnight sun periods are long and the daily cycle weak, such a strong daily signal in the mean number of observations is found.

P8L2: Here we are at the end of the description of the skin-temperature data treatment and there is nothing about treatment of data gaps introduced by cloud cover. Please correct me if I am wrong, but if I am right, that is a major omission. Given this absence, my presumption is - I cannot find any clarification in this manuscript - data gaps are left open; if one has no method to fill data gaps these gaps remain gaps. In favor of my presumption, Figure 6 has also regions with no data. Introducing gaps when your method fails positively bias your method performance and introduces an unknown bias in the final result. Again, please correct me if I am wrong; but if I am not, again, this data set cannot be used as all-weather T2m dataset and the paper cannot be accepted for publication until the cloud problem is resolved.
Furthermore, no comments are made in how sub-tile temperature variations due to topography are dealt. I thus presume it is ignored, fine, but state explicitly. It

does affect your correction procedure of 3.1.

P8L6: The discussion of the comparison with in situ observations is a missed chance. It allows you to understand why remote sensed skin temperatures are deviating. Does the correlation improve if the exercise is repeated using valid 3-hourly estimates? If so, then it is a data gap (= cloud) problem. Furthermore, as surface temperatures are very elevation dependent, this must be discussed as many of the PROMICE AWSs are close to the ice sheet margin, thus in terrain in which the elevation potentially varies more than 1000 m in a 0.25° degree tile.

P9L11: It is very common in comparable studies to cut your dataset into 3. In that case, you can perform the training-validation cycle three times; all three data subsets are used once in a training-validation cycle for validation and the remaining two are used for training in that cycle. Why is this approach not applied here?

P10: Equations 4 to 7 provide an elegant approach to evaluate rather simple correction functions, Eqs. 8-12. Still, this is not the best you can do. Why is not a state-of-the-art method like a neural-network approach used? Furthermore, as there are data gaps due to clouds, how are they dealt here? Are these days neglected?

P17F8: The paper gives me no real clue how these data gaps are filled.

As you can see, I am not convinced of the scientific soundness of the approach to convert clear-sky remote sensed skin temperatures into a continuous daily T2m data set. If I would have to do this, I would have taken the following approach: Take the discontinuous 3-hourly dataset of clear-sky skin temperatures, the continuous dataset of the cloudy/clear sky observation ratio, and other sensible remote sensed data products (like cloud properties, shortwave fluxes, atmospheric temperatures, sea ice state, local topography) and put it all into a neural network method (or any other AI) to order to produce a continuous 3-hourly T2m time series, trained with and evaluated

against the in situ dataset. In a final step the 3-hourly data are averaged to daily means.

I have read the results section as if the authors had produced a sensible daily T2m dataset, as it is possible that they indeed did this, but failed to convey that to me. At the other hand, since my objections to the method (description) are so major, it makes no sense to do detailed suggestions how the results, discussion and conclusions sections may be improved.

The results section analyses if there are systematic biases as function of the estimated T2m. In a renewed submission, the authors should analyze separately the performance for clear-sky, mixed sky and fully cloudy conditions. It should be proven that a reasonable method is found to estimate T2m for all conditions.

Table 8 and Figure 11 show in my humble view that the data set presented here is not good enough to be used. As e.g. Batrak and Müller (https://doi.org/10.1038/s41467-019-11975-3P) demonstrate, ERA-Interim and ERA5 and other reanalyses do a very poor job over the Arctic ocean due to missing snow cover over sea ice and misrepresented sea ice thickness. For Greenland, I have no paper at hand that does a similar analysis and I am neither aware that ERA-Interim is doing an extremely poor job there too. As you did not mention anything about applying an elevation correction on the reanalysis data – which is essential for a fair comparison, I suspect that overlook might be part of the poor performance of ERA-Interim over the ice sheet. Nonetheless, a useful t2m product derived from remote sensing should be able to beat easily a flawed model product – and yours does not.
Furthermore, as these reanalyzes fail to represent Arctic T2m, they should not be used as benchmark. The data set should be benchmarked against reanalysis results of RCMs optimized for either the Arctic or Greenland. I know there are several colleagues at your institute that can help you in selecting appropriate RCMs and retrieving the data.

Finally, the discussion leaves me puzzled by the fact that the authors are aware of the cloud-gap problem, but try to publish a data set in which this problem is not fully solved (as there are data gaps in the presented data set) and failed to present properly the measures they have undertaken to mitigate the "cloud problem".

---

## Referee Comment (RC2) · Anonymous Referee #2 · 23 Oct 2019

General: This study aims to produce a 2-m air temperature product for the Arctic region by establishing multiple linear regression models with satellite-derived surface temperature measurements and other covariates. The goal of this study is well stated, and the product is potentially very useful. However, the authors tend to provide over-whelming details on some data processing techniques that already exist in two previous papers while the most important issue on cloud influences are much less discussed. Some parts of methodology and validation may have some mis-interpretation and still needs careful discussions. I am suggesting the following major/minor comments for authors to revise their manuscript.

[Figure]

Major: It is a bit concerning to me that many methodology materials have already been presented in Nielsen-Englyst et al. (2019) and Hoyer et a. (2018), but the authors spent many pages describing the same amount of details in this paper. I would suggest the authors to only retain the most important information, and then delete all other repeating materials by citing those previous studies (authors should start from already pre-processed data, direct readers to read those two papers, and then leave more room for remaining story). I think it's not suitable to publish the same amount of details in this paper again, especially considering Nielsen-Englyst's other paper was also published in The Cryosphere.

P9 Methods: Why choose 0.5 degree as the matching threshold? I think this may be too large. If the authors reduce this threshold and reduce number of matching pairs, will the results be better? Please show some analyses, or at the very least, this needs to be carefully discussed.

P12L12: I think the highest correlation in ISTskinSeason may be wrong interpretation, because what the authors computed are raw data correlation. Without removing the seasonal cycle, this calculation of correlation will be dominated by seasonal cycle, and of course ISTskinSeason will have better correlation. Since the authors' goal is to produce a T2m product, this product should aim at achieving high accuracy in anomalous T2m days, and also being able to capture general characteristics such as seasonal cycle (for general analysis) and trend (for global warming analysis). In my opinion, I think the authors should aim at achieving the highest anomaly correlation when training their models, and revise relevant parts when interpreting model performance.

There seems to be overwhelming technical details provided, however, the most important issue is on how the cloud days were considered, but this seems to receive the least attention? Can the authors reduce some details as I mentioned in Comment #1, and leave more room for how the cloud days were treated? Those are the most useful and challenging issues for this product.
P1L16: again, if using this raw correlation, my guesses are that if you simply add a systematic bias (e.g., +0.3 degree C) to the satellite surface temperature to derive T2m, you will probably also get similar high correlation. Can authors perform this simple calculation and demonstrate more clearly on the gains of their regression model?

Minor: P2L26: change to "significantly different"

P7L4-5: I am not familiar with this (18-6, 6-18) way of presenting time. Is this following any convention? If not, I suggest the authors to make revisions.

P9L5: delete "the" before "different regions"

P9: Unclear presentation as to what is the percentage split for training/testing. Please revise presentation in the format of e.g., 80%/20% split.

Table 3: It would help if the authors can use stars to mark significance level of the correlation (e.g., *: $p<0.05$)... Plainly presenting the correlation is less informative

P11L7: Warming effects are mainly resulted from high clouds, but low clouds can cool the surface. Was this differentiated? Can authors provide some discussions on this?

P23L21: again, this highest correlation may not be a good indicator for this product to be reliable. Please see my earlier comments and provide some revisions or discussions on this matter.

Abstract: authors should mention their data product's spatial resolution in the abstract.

---

## Author Comment (AC1) · 8 Dec 2019

See attached pdf

Please also note the supplement to this comment:
https://www.the-cryosphere-discuss.net/tc-2019-126/tc-2019-126-AC1-
supplement.pdf
* * *

---

## Author Comment (AC2) · 8 Dec 2019

**General author response**

We would like to thank both reviewers for their time and efforts put into reviewing this manuscript. There are many good comments and questions, which can lead to a much stronger and clearer manuscript.

However, there are some misunderstandings that we would like to clarify. The paper presents a clear sky T2m product based on clear sky satellite observations, similar to the clear sky T2m products which have previously been derived over ocean, lakes and land from satellite observations (e.g. Good et al., 2016). Obviously, we have failed to address this clearly, since it is pointed out by both reviewers (even though in the abstract (P1L14), we do state: "The satellite derived T2m product including estimated uncertainties covers clear sky snow and ice surfaces in the Arctic region during the period 2000-2009").

Below is a summary list of why we believe the paper and the derived T2m dataset are highly relevant for publication:

- Infrared surface temperature retrievals are not possible in cloudy conditions. This is similar to all other infrared retrievals of e.g. Sea Surface Tempeature and Land Surface Temperature (for the other surface types). These clear sky observations are used extensively in models and aggregated products and are among the data sources that give the largest impact and improvement on model forecasts.
- Similar satellite based T2m relationship models have been derived for land surfaces with great success and large uptake (Good, et al., 2016) and the investigation of the ISTskin versus T2m relationships over ice and the derivation of T2m from ISTskin were identified in Merchant et al. (2013) as very important areas for improving the understanding of the surface temperature of the Earth.
- The number of satellite observations in the Arctic region is much higher than what is obtained from traditional in situ observations.
- The product described here is not meant to replace in situ observations or existing T2m products, but should be considered as complementary to existing observations.
- We have added results demonstrating that that we have a 94% daily average coverage throughout the years 2003-2009 of T2m, representing all-sky temperatures for the GrIS, considering a 1x1 degree grid. For the sea ice region, the same number is 81%. The coverage is stable from 2003-2009 and somewhat lower before 2003.The 2003-2009 period is chosen here to represent the coverage in the recent satellite observing constellation.
- The days when the satellite derived T2m product is available, it represents the all-sky T2m, since it has been regressed towards in situ measurements obtained both in cloudy and clear sky conditions (see discussion in manuscript).
- We recognize that filling the gaps due to clouds is an important topic, which is outside the scope of this paper. The starting point, however, before producing a gap filled product is the product we describe here.
- The work is part of a Horizon 2020 project EUSTACE (lead by UK Metoffice) and will feed into a new global T2m analysis similar to e.g. the CRUTEMP and GISS, but also including satellite information (Rayner et al., 2019). The data set we present here will thus improve the global T2m estimates, which are among the most used and cited climate data sets worldwide.
- A revised discussion has been added to the methodology section, clarifying that the seasonal signals in the T2m regression models are included in all the models (through the IST relation) and that the extra terms in equations 9-12 represent the anomalies, as suggested by reviewer #2. This has been clarified in the text

Based upon the arguments listed above and the clarifications in the revised version (see attached manuscript with track-changes below), it is our sincere hope that the reviewers will realize the large potential and value of the T2m data set we describe here. Below, we have responded to the reviewers' comments point by point.

**RC1 – Authors response**

*Author response is blue.

The authors present a remote sensed 2m temperature product for the Arctic ocean and the Greenland Ice sheet. For this aim, they use Arctic and Antarctic Ice Surface Temperatures from thermal Infrared satellite Sensors (AASTI) data set and apply a correction to convert surface temperatures into 2 m temperatures. The derived temperatures are compared with in situ observations and the data set performance is compared to the performance of T2m from ERA-Interim.

First of all, I apologize for the long time I needed to complete this review, which added to the long time that the editor needed to find reviewers. Sadly, I'm not convinced that the authors properly resolve all the challenges that occur when remote sensed surface temperatures are converted into t2m temperatures.

My largest objection is that the authors fail to properly resolve the "cloud problem". Infrared satellites can only measure surface temperatures in cloud free conditions. However, cloudy conditions lead totally different weather than cloud-free conditions, especially in winter. Too little attention is paid to this problem in the methodology and results section. A publishable data set must resolve the cloud-data-gap problem and it should be explicitly shown that reasonable estimates can be provided for cloudy conditions, even if these conditions last for days or more. As the authors do not aim to present a clear-sky T2m product, which would be of limited value, this shortcoming in the methodology and discussion must be resolved prior acceptance can be considered.
We acknowledge and regret that it has not been stated clearly enough that we derive a clear sky T2m product (see general author response). Days with clouds and few clear sky observations (as explained in detail at P7L4:14) are not considered in this analysis, as we do not have any or sufficient observations to provide an estimate of the daily IST (i.e. to resolve the diurnal cycle). Please see revised abstract, introduction, results, discussion and conclusion sections, which hopefully makes it clear that this product covers clear sky conditions only.

We do not agree that clear sky products are of limited value (see general author response and the introduction of the revised manuscript). We believe that it is of great value to establish a relationship between surface air temperature measurements and satellite-based estimates of the ice surface temperature, with the aim of estimating surface air temperature in regions where no in situ observations are available (and thus drastically increasing the density of surface air temperature information globally). The derived T2m_sat product including well-characterized uncertainties can improve existing reanalysis products in regions with limited in situ observations (such as the Arctic), and thus provide more complete temperature fields.

Specific major and minor comments:
P3L13: which "data"? I presume satellite data, as AWSs do not move. This must be clearly worded.
Unfortunately, we cannot identify what the reviewer is referring to here. The in situ data is described in Section 2.1 and the satellite data in Section 2.2.

P4L10: Snow on sea ice have a major effect on the measured temperature as snow is a very good insulator (e.g. Graham et al, 2019, https://doi.org/10.1038/s41598-019-45574-5). Hence, if the buoy thermistor has a smaller diurnal cycle as the T2m sensor, snow cover is affecting the observation and buoy thermistor should be discarded as valid surface temperature observation. Unless "unrealistic data artifacts" includes damped daily cycles - which then should be stated explicitly -, I believe data scarcity should not be an excuse for retaining incorrect data.

We agree on this and we have now removed these data from the analysis. The variability method can only work for periods where significant diurnal variability is present and is thus not a robust method. No data of from these types of observations are included in the results. Thank you for pointing this out.

P5L4: All three citations listed to introduce the AASTI refer to technical documents; they do not refer to peer reviewed papers. This is not extremely relevant in itself, but it raises, in my humble opinion, the necessity that the authors restate briefly the methods to compile this dataset. Furthermore, from the title of Høyer 2019, this dataset is only providing clear-sky ice surface temperatures. This must be restated when the AASTI dataset is introduced.
Thanks, we agree on this. "Clear sky" has been added in the text, where the AASTI dataset is introduced. Furthermore, a brief description of the AASTI IST algorithm has been added following your advice.

P6L10: It does not become clear to me how these 3-hourly bin averages are aggregated into one daily value. The procedure should be added and described plainly.

The 3-hourly bin averages are not aggregated into one daily value. All the available observations within a day and 0.25 degrees are aggregated to produce the daily estimates. And similarly for the 3-hourly averages. The 3-hourly bin averages are only used to estimate the satellite sampling during the day and to gain confidence in the daily cycle estimates (as stated in P6L14-15).

We have reformulated this part to make it more clear. Also, we have expanded on the description of the AASTI data (and algorithm) to make it more clear that the original AASTI data is swath based (L2).

P7F3: Although the figure is somewhat instructive, I would be more interested to see (also) **a)** the ratio between cloudy and cloud free observations, as the current figure is clouded by the variations in observation density and **b)** the percentage of days with one or more valid observations within every time interval. Increasing from 1 to 25 observations in a 3-hour interval improves the measurement accuracy, decreasing from 1 to 0 leads to a data gap. By the way, I am puzzled by the fact that even far North (>75 N), where the polar night and midnight sun periods are long and the daily cycle weak, such a strong daily signal in the mean number of observations is found.

[Figure]

We agree with the reviewer that it is a good idea to provide some information on the coverage of the T2m$_{sat}$ product (and thus the days where the T2msat is not available due to clouds). This information has now been added to the manuscript in the discussion and conclusion. The coverage is stable from 2003-2009 and somewhat lower before 2003 (see figure above). The average daily coverage is 84% and 67% for land ice and sea ice, respectively, considering the stable 2003-2009 period and the 0.25 degree grid. When considering a 1 degree grid resolution these numbers increase to 94% and 81%, respectively. The 2003-2009 period is chosen here to represent the coverage in the recent satellite observing constellation. The high percentages in coverage demonstrate that the gaps due to cloudy days are limited and that the data set contains a significant amount of information on the all-sky daily T2m even though it is based upon clear sky satellite observations.

Note that the daily cycle in the satellite sampling depends both on the daily variations in cloud cover and the quality of the cloud screening, which is significantly improved for day light conditions. Also northwards of 80 degrees there is little daily variations, as expected.

P8L2: Here we are at the end of the description of the skin-temperature data treatment and there is nothing about treatment of data gaps introduced by cloud cover. Please correct me if I am wrong, but if I am right, that is a major omission. Given this absence, my presumption is - I cannot find any clarification in this manuscript - data gaps are left open; if one has no method to fill data gaps these gaps remain gaps. In favor of my presumption, Figure 6 has also regions with no data. Introducing gaps when your method fails positively bias your method performance and introduces an unknown bias in the final result. Again, please correct me if I am wrong; but if I am not, again, this data set cannot be used as all-weather T2m dataset and the paper cannot be accepted for publication until the cloud problem is resolved.

You are right that days with clouds (as defined in the paragraph P6L10-P7L14) do not contain any T2m estimates (see general author response). That is indeed the reason you see gaps in the example figure (Figure 6). However, this is similar to all other infrared retrievals of e.g. Sea Surface Temperature or Land Surface Temperature (for the other domains), and these clear sky observations are used extensively in models and aggregated products with large positive impacts on model forecasts.

Furthermore, no comments are made in how sub-tile temperature variations due to topography are dealt. I thus presume it is ignored, fine, but state explicitly. It does affect your correction procedure of 3.1.

Thanks for pointing this out. This has now been stated explicitly and discussed in section 5.

P8L6: The discussion of the comparison with in situ observations is a missed chance. It allows you to understand why remote sensed skin temperatures are deviating. Does the correlation improve if the exercise is repeated using valid 3-hourly estimates? If so, then it is a data gap (= cloud) problem.

We know from other validation studies, that the largest component of uncertainty in the satellite retrievals is typically the influence of undetected clouds. Clouds are very hard to detect over sea ice and will typically introduce a cold bias.

Repeating the exercise with the aggregated 3-hour IST_L3_skins will give us all-sky 3-hourly estimates of T2m. The aim of this paper is to derive a daily all-sky values and not 3-hourly values. However, a validation of 3-hourly T2m will likely be improved (compared to the daily values), since each 3-hour value will be validated against in situ observations taken within 3 hours from the satellite measurement. For daily estimates, we allow (only) two 3-hour bins (night/day) with available satellite observations in each day and this may result in larger differences, if these are not representative of the entire day. However, this effect has been mitigated by the checks listed in Section 2.2.

One way to investigate the effect from data gaps due to clouds (within the derived daily T2m product) is to look at the performance against the number of filled 3-hour bins (empty 3-hour bins means that clouds are present). The number of filled 3-hour bins increase over the period (see Figure 8) due to the increasing amount of satellites (see Figure 2). However, we do not see an increase in the performance of the estimated T2m over the period (Figure 7), which means that more filled 3 hour bins not necessarily improve the performance of our product (but possible improve the coverage). This has been clarified in manuscript.

Furthermore, as surface temperatures are very elevation dependent, this must be discussed as many of the PROMICE AWSs are close to the ice sheet margin, thus in terrain in which the elevation potentially varies more than 1000 m in a 0.25_ degree tile.

We agree that the elevation change within the matchup distance will introduce differences. The average slope of the 8 PROMICE stations is 1.49°, and this (together with the matching distance) increase the uncertainty in a pixel-to-point-measurement comparison, but as discussed in RC-2 P9 the matching distance has been set to obtain a robust number of in situ observations. Thanks for pointing this out. It has been stated now explicitly in the discussion.

P9L11: It is very common in comparable studies to cut your dataset into 3. In that case, you can perform the training-validation cycle three times; all three data subsets are used once in a training-validation cycle for validation and the remaining two are used for training in that cycle. Why is this approach not applied here?

We believe that the method used here, where we divide the data into two independent subsets (one for training and one for validation) is a suitable approach for this problem, which gives us realistic and independent information about the performance of the regression method. The limited amount of in situ observations makes it critical to divide the data into even more subsets.

P10: Equations 4 to 7 provide an elegant approach to evaluate rather simple correction functions, Eqs. 8-12. Still, this is not the best you can do. Why is not a state-ofthe-art method like a neural-network approach used? Furthermore, as there are data gaps due to clouds, how are they dealt here? Are these days neglected?

Days with clouds are neglected (see general comment and comment to P8L2). We agree that the use of advanced statistical methods to fill the gaps is very interesting and in the discussion we state that in order to fill gaps (due to cloudy conditions) different methods can be used such as a statistical technique or the use of atmospheric models and assimilation (P22L8). However, we believe that deriving a gap-free T2m product is outside the scope of this paper.

P17F8: The paper gives me no real clue how these data gaps are filled. As you can see, I am not convinced of the scientific soundness of the approach to convert clear-sky remote sensed skin temperatures into a continuous daily T2m data set. If I would have to do this, I would have taken the following approach: Take the discontinuous 3-hourly dataset of clear-sky skin temperatures, the continuous dataset of the cloudy/clear sky observation ratio, and other sensible remote sensed data products (like cloud properties, shortwave fluxes, atmospheric temperatures, sea ice state, local topography) and put it all into a neural network method (or any other AI) to order to produce a continuous 3-hourly T2m time series, trained with and evaluated against the in situ dataset. In a final step the 3-hourly data are averaged to daily means.

See general comment and comment to P8L2. As mentioned in P8L6, repeating the exercise with the aggregated 3-hourly satellite ISTskins, would result in an all-sky 3-hourly T2m estimate. Averaging the all-sky 3-hourly T2m estimates will lead to a (unknown) clear-sky bias in the daily averaged value (in cases

where only a few 3-hour bins are filled within a day), and it cannot be used as an all-weather daily T2m estimate (in contrast to what we derive here, by tuning to daily all-sky in situ T2m).

I have read the results section as if the authors had produced a sensible daily T2m dataset, as it is possible that they indeed did this, but failed to convey that to me. At the other hand, since my objections to the method (description) are so major, it makes no sense to do detailed suggestions how the results, discussion and conclusions sections may be improved.

We hope that the reviewer will read and provide suggestions to the discussion and conclusions in the revised version. We have changed the discussion and conclusions to take into account the general comments from both reviewers and believe this has led to improved sections.

The results section analyses if there are systematic biases as function of the estimated T2m. In a renewed submission, the authors should analyze separately the performance for clear-sky, mixed sky and fully cloudy conditions. It should be proven that a reasonable method is found to estimate T2m for all conditions.

See also general comment and comment to P8L2. The days when the satellite derived T2m product is available, it represents the all-sky T2m average for that day, since it has been regressed towards in situ measurements obtained both in cloudy and clear sky conditions. This has been clarified in manuscript.

Table 8 and Figure 11 show in my humble view that the data set presented here is not good enough to be used. As e.g. Batrak and Müller (https://doi.org/10.1038/s41467-019-11975-3P) demonstrate, ERA-Interim and ERA5 and other reanalyses do a very poor job over the Arctic ocean due to missing snow cover over sea ice and misrepresented sea ice thickness. For Greenland, I have no paper at hand that does a similar analysis and I am neither aware that ERA-Interim is doing an extremely poor job there too. As you did not mention anything about applying an elevation correction on the reanalysis data – which is essential for a fair comparison, I suspect that overlook might be part of the poor performance of ERA-Interim over the ice sheet.
Nonetheless, a useful t2m product derived from remote sensing should be able to beat easily a flawed model product – and yours does not.
Furthermore, as these reanalyzes fail to represent Arctic T2m, they should not be used as benchmark. The data set should be benchmarked against reanalysis results of RCMs optimized for either the Arctic or Greenland. I know there are several colleagues at your institute that can help you in selecting appropriate RCMs and retrieving the data.

It is indeed a fact that current reanalyses do a very poor job in the Arctic. This is one of the main reasons that using satellite observations to provide an alternative and independent estimate of the near surface air temperatures is of great value in particular for the Arctic region. This has now been clarified in the introduction.
We think the comparison against ERA-I is relevant and fair, since no elevation correction is applied to the satellite derived T2m product either. Part of the discrepancy between the in situ observations and the T2m estimates (both ERA-I and the satellite derived T2m) is the point to pixel comparison, where the point measurement may not be representative of the entire pixel. The differences here arise indeed both because of differences in elevation but also due to the matchup distance. Even though the satellite derived products beats ERA-I over the Greenland Ice Sheets, the most important take-away message is what is (as stated in the conclusion P24L4) that the derived T2m product should not replace the already existing air temperature measurements, but rather to supplement these e.g. in areas where no in situ observations are currently available. Since, the errors in ERA-I and the $T2m_{sat}$ are independent and uncorrelated a combination of the two datasets would lead to an even better T2m estimate. We have clarified this when the comparison against ERA-I is performed.

ERA-I is one of the most used reanalysis products and the T2m has been used extensively for many publications, also in the Arctic. It is therefore of general interest to compare the performance against this product. ERA-5 and C3S Arctic regional reanalysis were not available at time of the analysis but will be considered for future work.

Finally, the discussion leaves me puzzled by the fact that the authors are aware of the cloud-gap problem, but try to publish a data set in which this problem is not fully solved (as there are data gaps in the presented data set) and failed to present properly the measures they have undertaken to mitigate the "cloud problem".

See general author response.

**RC2 – Authors response**

General: This study aims to produce a 2-m air temperature product for the Arctic region by establishing multiple linear regression models with satellite-derived surface temperature measurements and other covariates. The goal of this study is well stated, and the product is potentially very useful.

We are glad to hear that the reviewer acknowledge that the product potentially is very useful. We hope that the comments below and the changes in the manuscript resolve the issues that the reviewer has pointed out.

However, the authors tend to provide over-whelming details on some data processing techniques that already exist in two previous papers while the most important issue on cloud influences are much less discussed. Some parts of methodology and validation may have some mis-interpretation and still needs careful discussions. I am suggesting the following major/minor comments for authors to revise their manuscript.

Point-by-point response is given below.

Major: It is a bit concerning to me that many methodology materials have already been presented in Nielsen-Englyst et al. (2019) and Hoyer et a. (2018), but the authors spent many pages describing the same amount of details in this paper. I would suggest the authors to only retain the most important information, and then delete all other repeating materials by citing those previous studies (authors should start from already pre-processed data, direct readers to read those two papers, and then leave more room for remaining story). I think it's not suitable to publish the same amount of details in this paper again, especially considering Nielsen-Englyst's other paper was also published in The Cryosphere.

The methodology section has been shortened as suggested, and in particular the repetitions of the Nielsen-Englyst et al. (2019) results have been reduced. We do keep most of the information from Hoyer et al. (2018), since this is not a peer reviewed paper, following the arguments of reviewer #1 (P5L4).

P9 Methods: Why choose 0.5 degree as the matching threshold? I think this may be too large. If the authors reduce this threshold and reduce number of matching pairs, will the results be better? Please show some analyses, or at the very least, this needs to be carefully discussed.

The matching threshold was chosen to ensure a sufficient number of matchups to derive and validate the different regression models with sufficient statistical confidence. This has been added now, when it is introduced. We do mention in the discussion that the limited number of matching pairs is one of the greatest challenges in this study (and similar studies). This part has now been modified and extended to explicitly mention the matching threshold and elevation effects as the main limitations in the derivation and validation of the regression model. Thanks for pointing this out.

P12L12: I think the highest correlation in ISTskinSeason may be wrong interpretation, because what the authors computed are raw data correlation. Without removing the seasonal cycle, this calculation of correlation will be dominated by seasonal cycle, and of course ISTskinSeason will have better correlation. Since the authors' goal is to produce a T2m product, this product should aim at achieving high accuracy in anomalous T2m days, and also being able to capture general characteristics such as seasonal cycle (for general analysis) and trend (for global warming analysis). In my opinion, I think the authors should aim at achieving the highest anomaly correlation when training their models, and revise relevant parts when interpreting model performance.

We agree with the reviewer that it is important to consider the strong seasonal cycle existing in the T2m. All the regression models we test here include a linear relation between the Tskin and T2m and thus all of them capture the large seasonal cycle in Tskin (through the $\alpha_1$ term). The close coupling between the Tskin and T2m on (synoptic (2-5 days) and seasonal time scales) is the main reason why we generally get very high correlations for all the regression models tested. The high correlations are thus expected and indeed dominated by the seasonal cycle in both temperatures. The following sentence has been added in the IST$_{skin-L3}$ validation in Section 2.2.: "The generally high correlations are dominated by the synoptic (2-5 days) and seasonal variations, which are pronounced in both IST and T2m."

The regression models listed in Eq.9 through Eq. 12 are included to examine how to best represent the residual variability not related to the synoptic and seasonal variations, as none of these terms depend upon the Tskin. To our understanding, this is actually what the reviewer asks for. As evident from Table 4, the ability to best represent the anomalous T2m days are actually found by including the seasonal component that only works on correcting the T2m-Tskin differences (and is not related to the Tskin itself, as these are represented by the $\alpha_1$ term). Table 4 shows that we get an improved correlation and RMS by using this model compared to the other. This discussion and clarification has been clarified in the text (see Section 3.1) and in conclusion.

There seems to be overwhelming technical details provided, however, the most important issue is on how the cloud days were considered, but this seems to receive the least attention? Can the authors reduce some details as I mentioned in Comment #1, and leave more room for how the cloud days were treated? Those are the most useful and challenging issues for this product.

The technical details have been reduced as suggested. Days with clouds (or days where we do not have observations in both the night/day bin as explained in detail in the paragraph P6L10-P7L14) are not considered in this analysis, as we do not have sufficient observations to provide an estimate of the daily IST. The final product only provides at T2m estimate for days where daily ISTs are estimated in respect to the criteria presented in P6L10-P7L14. The days when the satellite derived T2m product is available, it represents the all-sky T2m, since it has been regressed towards in situ measurements obtained both in cloudy and clear sky conditions. Please see changes in abstract, introduction, results, discussion and conclusion sections, which hopefully makes it clear that this product covers clear sky conditions only (e.g. when satellite observations are available).

P1L16: again, if using this raw correlation, my guesses are that if you simply add a systematic bias (e.g., +0.3 degree C) to the satellite surface temperature to derive T2m, you will probably also get similar high correlation. Can authors perform this simple calculation and demonstrate more clearly on the gains of their regression model?

(This question is related to P12L12)
Yes, this is exactly what we have shown in first row in Table 4 (from Eq. 8). The first regression model simply adds a systematic bias and a scaling factor of the IST and we do get an improved RMS and correlation (see Table 4) compared to using the IST alone (see Table 2). Table 4 also shows the results of using different predictors (including the seasonal cycle). Only, the inclusion of the SWd radiation and the seasonal cycle shows significant improvements from using a simple correction as suggested by the reviewer.

Minor: P2L26: change to "significantly different"
Implemented. Thanks!

P7L4-5: I am not familiar with this (18-6, 6-18) way of presenting time. Is this following any convention? If not, I suggest the authors to make revisions.
This has been rewritten in the manuscript to: "In order to best resolve the diurnal cycle with satellite information we require data during both night (between 18 and 6 local solar time) and day (between 6 and 18 local solar time) in order to calculate ISTskin_L3."
This is following the mathematical notation and terminology in the manuscript preparation site on the TC.net, date and time (https://www.the-cryosphere.net/for_authors/manuscript_preparation.html).

P9L5: delete "the" before "different regions"
Implemented

P9: Unclear presentation as to what is the percentage split for training/testing. Please revise presentation in the format of e.g., 80%/20% split.
The information has been added. Thanks.

Table 3: It would help if the authors can use stars to mark significance level of the correlation (e.g., *: p<0.05). . . Plainly presenting the correlation is less informative
This table has been removed in response to the reviewer's major/first comment.

P11L7: Warming effects are mainly resulted from high clouds, but low clouds can cool the surface. Was this differentiated? Can authors provide some discussions on this?
In Nielsen-Englyst et al. (2019), there was not differentiated between low/high clouds. But overall it was found that the Tskin increased in cloudy conditions compared to clear sky conditions. This is in agreement with Intrieri (2002), who found that clouds tend to have an overall warming effect in the Arctic (considering and discussing both the warming effect from high clouds and the cooling effect from low clouds). We refer to Intrieri (2002) for further discussion on this topic. We have left out this part (repeated from Nielsen-Englyst et al., 2019), since it is not crucial information in this paper (in response to reviewer's general comment to the methodology section), where we do not consider longer periods (>1 day) with clouds (see general comment). In this paper, the different cloud types are not important, as there are no satellite observations of the surface in any case.

P23L21: again, this highest correlation may not be a good indicator for this product to be reliable. Please see my earlier comments and provide some revisions or discussions on this matter.
Please see our responses to P12L12 and P1L16. We have added "and the lowest RMS" here.

Abstract: authors should mention their data product's spatial resolution in the abstract.

We agree on this and the information has been added.

**Deriving Arctic 2 m air temperatures over snow and ice from satellite surface temperature measurements**

Pia Nielsen-Englyst[1,2], Jacob L. Høyer[2], Kristine S. Madsen[2], Rasmus T. Tonboe[2] and Gorm Dybkjær[2]

[1] Technical University of Denmark (DTU), DK-2800 Kongens Lyngby, Denmark
5 [2] Danish Meteorological Institute (DMI), DK-2100 Copenhagen Ø, Denmark

*Correspondence to*: Pia Nielsen-Englyst (pne@dmi.dk)

**Abstract.**

The Arctic region is responding heavily to climate change, and yet, the air temperature of Arctic, ice covered areas is heavily under-sampled when it comes to in situ measurements, and large uncertainties exist in weather- and reanalysis products. This
10 paper presents a method for estimating daily mean clear sky 2 meter air temperatures (T2m) in the Arctic from satellite observations of skin temperature, using the Arctic and Antarctic ice Surface Temperatures from thermal Infrared (AASTI) satellite dataset, providing spatially detailed observations of the Arctic. The method is based on a linear regression model, which has been developed usingtuned against in situ observations to estimate daily T2m based on and daily mean clear sky satellite ice surface skin temperatures combined with a seasonal variation to estimate daily T2m. The daily satellite derived
15 T2m product including estimated uncertainties covers clear sky snow and ice surfaces in the Arctic region during the period 2000-2009, estimated on a 0.25 degree regular latitude-longitude grid. Comparisons with independent in situ measured T2m gives average correlations of 95.5% and 96.5% and average root mean square errors of 3.47°C and 3.20°C for land ice and sea ice, respectively. The reconstruction provides a much better spatial coverage than the sparse in situ observations of T2m in the Arctic, is independent of numerical weather prediction model input and it therefore provides an important alternative
20 to simulated air temperatures to be used for assimilation or global surface temperature reconstructions. A comparison between in situ T2m versus T2m from satellite and ERA-Interim shows that the T2m derived from satellite observations validate similar or better than ERA-Interim estimates in the Arctic.

**1 Introduction**

The Arctic climate is changing rapidly with surface temperatures rising faster than other regions of the world due to Arctic
25 amplification (Graversen et al., 2008; Pithan and Mauritsen, 2014). Meteorological measurements show that the 2000s were the warmest decade in Greenland since meteorological measurements started in the 1780s (Box et al., 2019; Cappelen, 2016; Masson-Delmotte et al., 2012).

The Arctic surface air temperature is one of the key an important climate indicators used to assess of regional and global climate changes (Hansen et al., 2010; Pielke et al., 2007) and both as most model simulations and observations by global

 indicate that  warming in the global climate be amplified at the northern high latitudes (e.g. Collins et al., 2013; Holland and Bitz, 2003; Overland et al., 2018). Traditionally, near surface air temperatures have been measured at the height of 1-2 m using automatic weather stations (AWSs) or buoys (Hansen et al., 2010; Jones et al., 2012; Rayner, 2003; World Meteorological Organization, 2014). Extreme temperatures, winds and the remoteness of the Arctic make in situ observations in the Arctic temporally and spatially sparse (Reeves Eyre and Zeng, 2017), and challenging. In particular, it is difficult to achieve climate-quality temperature records for this region.

The global near surface air temperature datasets that are currently most widely used (HadCRUT4, NOAAGlobalTemp and GISTEMP) are derived only by using in situ observations (Hansen et al., 2010; Morice et al., 2012; Smith et al., 2008; Vose et al., 2012). To increase the coverage and quality of the surface temperature products, polar orbiting satellites can offer a very good supplement to the in situ observations through a high spatial and temporal coverage of all regions in the Arctic.  Daily near surface air temperatures derived from satellites therefore have the potential to increase the amount of information in the data sets and improve the quality of the climate records, as recognized in Merchant et al. (2013) and Rayner et al. (2019).

 However, satellites with infrared sensors in the atmospheric window region of 10-12 micron wavelength measure the ice surface skin temperature ($IST_{skin}$) during clear skies whereas the current global temperature products estimate the near surface air temperature as are measured AWSs and buoys.  The surface skin temperature may differ considerably from the surface air temperature during melting conditions, but during other conditions the skin and surface air temperature may be more or less the same (Nielsen-Englyst et al., 2019).

To benefit from the good coverage of satellite surface temperature data, we have explored the relationships between the surface air temperature and the satellite measurements. Several studies have compared satellite retrieved $IST_{skin}$ and T2m from AWSs located on the Greenland Ice Sheet (GrIS; Dybkjær et al., 2012a; Hall et al., 2008, 2012; Koenig and Hall, 2010; Shuman et al., 2014) and over the Arctic sea ice (Dybkjær et al., 2012) and found temperature differences of which a significant part could be attributed to the temperature difference between T2m and $IST_{skin}$. Previously, work has been done to investigate the relationship between the surface and near-surface air temperature over ice using in situ observations (Adolph et al., 2018; Hall et al., 2008, 2004; Hudson and Brandt, 2005; Nielsen-Englyst et al., 2019; Vihma et al., 2008). Nielsen-Englyst et al. (2019) found that on average T2m is 0.65-2.65°C warmer than $IST_{skin}$ with variations depending on location of the measurement i.e. in the lower ablation zone, upper-middle ablation zone, accumulation zone, seasonal snow cover and sea ice. The T2m-$IST_{skin}$ difference was found to vary with season with smallest differences around noon and early afternoon during spring, fall and summer during non-melting conditions. Furthermore, wind speed and cloud cover were identified as key parameters determining the T2m- $IST_{skin}$ difference.

[revised manuscript text omitted]

The different in situ types measure the air temperature at different heights that furthermore differ over time depending on the amount of snow fall, snow drift and snow melt. Here, we will refer to T2m for all observation types regardless of these variations. Nielsen-Englyst et al. (2019) showed small changes (<0.22°C) in T2m-IST$_{skin}$ differences when using only observations within the measurement range of 1.90-2.10 m in height compared to using all measurements (ranging in measurement height from 0.3 m to 3 m). The accuracy of the air temperature sensors for all observation sites is approximated to 0.1°C (Hall et al., 2008; Høyer et al., 2017b). Few data sources provide both skin and air temperatures e.g. the PROMICE and ARM stations . The PROMICE skin temperatures have been calculated from up-welling longwave radiation, measured by Kipp & Zonen CNR1 or CNR4 radiometer, assuming a surface longwave emissivity of 0.97 (van As, 2011).

temperature becomes an internal snow temperature. However, in this analysis the buoy temperature measurements will be treated and counted as $IST_{skin}$ measurements, as we have no information on the snow depth on top of the buoys. All in situ data have been screened for spikes and other unrealistic data artefacts by visual inspection. Afterwards, the in situ observations have been averaged to daily temperatures using all available observations. Figure 1 shows the number of daily averaged in situ observations each year during 2000-2009 of $IST_{skin}$ and T2m over Arctic land ice and sea ice, respectively. In total 65,810 observations with daily T2m and 7,681 057 observations with daily $IST_{skin}$ are available over land ice. However, only 624 of these cover Arctic sea ice. See Table 1 for more information on the in situ observations used in this study.

[Figure]

**Figure 1: Total number of daily averaged in situ observations of T2m and $IST_{skin}$ over Arctic land ice and sea ice per year covering the period 2000-2009.**

**Table 1: Overview of in situ observations used in this study, covering 2000-2009.**

| | No. of sites, (AWS, buoys or ships) | No. of days with observations | Surface Type | Observation Type | Temperature measurements |
|---|---|---|---|---|---|
| ACSYS | 7 | 280 | Sea ice | Buoy | T2m |
| ARM | 2 | 2,846 | Land snow | AWS | T2m, IST$_{skin}$ |
| CRREL | 10 | 1,031 | Sea ice | Buoy | T2m |
| DAMOCLES | 25 | 2,160 | Sea ice | Buoy | T2m |
| ECMWF | 196 | 27,235 | Sea ice | Buoy | T2m,  |
| FRAMZY | 11 | 251 | Sea ice | Buoy | T2m |
| GC-NET | 15 | 29,133 | Land ice | AWS | T2m |
| POLARSTERN | 1 | 189 | Sea ice | Ship | T2m |
| PROMICE | 8 | 2,685 | Land ice | AWS | T2m, IST$_{skin}$ |

**2.2 Satellite data**

The satellite data used in this study is from the Arctic and Antarctic Ice Surface Temperatures from thermal Infrared satellite sensors (AASTI; Dybkjaer et al., 2018; Dybkjær et al., 2014; Høyer et al., 2019) data set, covering high latitude seas, sea ice, and ice cap clear sky surface temperatures based on satellite infrared measurements from the CLARA-A1 data set compiled by EUMETSAT's Climate Monitoring, Satellite Application Facility (Karlsson et al., 2013). The data set is based on one of the longest existing satellite records from the Advanced Very High Resolution Radiometer (AVHRR) instruments on board a long series of NOAA satellites. AASTI contains swath based (i.e. Level 2; (L2) ice surface skin temperature (IST$_{skin\_L2}$) data processed and error corrected on the original Global Area Coverage (GAC) grid. The first version of the AASTI product, used in this study, is available from 2000 to 2009 in the original projection and resolution (L2), i.e. ~0.05 arc degree resolution and multiply daily coverage. Since 2000, seven different AVHRR instruments have been orbiting the globe, each 14 times per day, and thus providing approximately bi-hourly coverage of the Polar Regions (Figure 2). The number of operational satellites has increased from 2 to 6 from 2000 to 2009. The IST algorithm used to generate in the AASTI data set is based on thermal infrared brightness temperatures of AVHRR channel 4 (centre wavelength at ~11 microns) and 5 (centre wavelength at ~12 microns), and the satellite zenith angle. The algorithm is a split window algorithm, working within three temperature domains for each individual satellite (Key et al., 1997). The retrieval calibration of each domain has been done by relating modelled surface temperatures with modelled top-of-atmosphere brightness temperatures, determined by a radiative transfer model (Dybkjær et al., 2014). Cloud masking has been performed using the Polar Platform System (PPS) processing cloud processing software (Dybbroe et al., 2005a, 2005b).

[revised manuscript text omitted]

$$G^{-g} = (G^T G + \varepsilon^2 I)^{-1} G^T \qquad (6)$$

$$m = G^{-g} d^{obs}, \qquad (7)$$

where $G^{-g}$ is called the generalized inverse, $\varepsilon$ is a damping factor and $I$ is an identity matrix (with ones in the diagonal and zeros elsewhere). The superscript operator T denotes transposing and -1 denotes inversion. We have tested a range of damping factors to assess the relation to the error coefficients. A damping factor of 0.2 was chosen to avoid overfitting noise in the data, while keeping the error coefficients low.

The choice of predictors is based on current knowledge of the parameters that influence the relationship between $IST_{skin}$ and $T2m_{insitu}$ (Nielsen-Englyst et al., 2019), limited by the available satellite data. Nielsen-Englyst et al. (2019) showed that  Tthe T2m-Tskin difference varies over the  season with smallest differences  during spring, fall and summer in non-melting conditions.

For that reason, we have also tested the effect of including a seasonal cycle as predictor. Nielsen Englyst et al. (2019) also found that at the observation sites located on the Arctic sea ice and snow covered regions of North Alaska the $T2m_{insitu}$-$IST_{skin}$ difference decreases almost linearly as a function of wind speed due to increased turbulent mixing of the air for higher wind speeds. Contrary, the maximum $T2m_{insitu}$-$IST_{skin}$ differences over the GrIS occur at wind speeds of about 5 m s$^{-1}$.

5  This is also seen by Adolph et al. (2018) at Summit, GrIS and by Hudson and Brandt (2005) at the South Pole, and the feature is related to the pronounced katabatic winds in these regions. Furthermore, Nielsen Englyst et al. (2019) found that the $T2m_{insitu}$-$IST_{skin}$ difference tends to decrease linearly as a function of the cloud cover fraction for all seasons and all regions. The reason for this is that clouds have a predominately warming effect on the skin temperature in the Arctic (Intrieri, 2002; Walsh and Chapman, 1998). Nielsen Englyst et al. (2019) showed an almost linear relationship between the

10  $T2m_{insitu}$-$IST_{skin}$ difference and the $IST_{skin}$, with larger differences for colder skin temperatures. Based on these findings we have calculated the correlations between satellite skin temperature ($IST_{skin\_L3}$), in situ surface air temperatures ($T2m_{insitu}$), latitude (Lat), downward shortwave radiation (SWd) and not considering clouds (theoretical), and wind speed (WS) from ERA Interim reanalysis. Since the cloud cover fraction and longwave radiation are unknown in this case, we have tested $IST_{skin\_L3}$ as a predictor instead. The resulting correlations are shown in Table 3.

15  **Table 3 Correlations between satellite-measured $IST_{skin\_L3}$, in situ measured $T2m_{insitu}$, latitude (Lat), theoretical downward shortwave radiation (SWd), and ERA-Interim wind speed (WS).**

|  |  | $IST_{skin\_L3}$ | $T2m_{insitu}$ | Lat | SWd | WS |
|---|---|---|---|---|---|---|
| | $IST_{skin\_L3}$ | 1.00 | 0.96 | -0.22 | 0.72 | -0.25 |
| | $T2m_{insitu}$ | 0.96 | 1.00 | -0.25 | 0.61 | -0.28 |
| Land ice | Lat | -0.22 | -0.25 | 1.00 | -0.05 | 0.10 |
| | SWd | 0.72 | 0.61 | -0.05 | 1.00 | -0.23 |
| | WS | -0.25 | -0.28 | 0.10 | -0.23 | 1.00 |
| | $IST_{skin\_L3}$ | 1.00 | 0.96 | -0.07 | 0.79 | -0.06 |
| | $T2m_{insitu}$ | 0.96 | 1.00 | -0.03 | 0.74 | -0.07 |
| Sea ice | Lat | -0.07 | -0.03 | 1.00 | 0.03 | -0.04 |
| | SWd | 0.79 | 0.74 | 0.03 | 1.00 | -0.21 |
| | WS | -0.06 | -0.07 | -0.04 | -0.21 | 1.00 |

The $IST_{skin\_L3}$ and air temperatures are well correlated (above 90% correlation), and $IST_{skin\_L3}$ also show correlation with the shortwave radiation. Part of the correlation between temperature and the theoretical shortwave radiation is expected to be

20  due to correlation of a seasonal cycle in both signals, not necessarily indicating causality. Therefore, for the regression modelling, a seasonal cycle with fit of amplitude and phase was also tested. A total of 5 regression models with different predictors have been tested (Høyer et al., 2018):

$\hat{\text{IST}}_{\text{skin}}$: $\qquad T2m_{sat} = \alpha_0 + \alpha_1 IST_{skin\_L3}$ (8)

$\hat{\text{IST}}_{\text{skin}}$SWd: $\qquad T2m_{sat} = \alpha_0 + \alpha_1 IST_{skin\_L3} + \alpha_2 SWd$ (9)

$\hat{\text{IST}}_{\text{skin}}$WS: $\qquad T2m_{sat} = \alpha_0 + \alpha_1 IST_{skin\_L3} + \alpha_2 WS$ (10)

$\hat{\text{IST}}_{\text{skin}}$Lat: $\qquad T2m_{sat} = \alpha_0 + \alpha_1 IST_{skin\_L3} + \alpha_2 Lat$ (11)

$\hat{\text{IST}}_{\text{skin}}$Season: $\qquad T2m_{sat} = \alpha_0 + \alpha_1 IST_{skin\_L3} + \alpha_2 \text{COS}\left(\frac{t \cdot 2\pi}{1\,yr}\right) + \alpha_3 \text{SIN}\left(\frac{t \cdot 2\pi}{1\,yr}\right)$ (12)

The regression model in Eq. (8) is limited to an offset and a scaling of IST$_{\text{skin\_L3}}$, where the latter term accounts for the synoptic and seasonal variations, which are the dominating factors in both IST and T2m variability. This part is thus included in all regression models tested. The while all other regression models also have a third predictor, which is included to examine how to best represent the residual variations in the T2m-IST difference. 
[revised manuscript text omitted]
., 2019). Days with clouds and few clear sky observations (as explained in Section 2.2) are not included in the dataset. However, for those days when the satellite derived T2m product is available, it provides an estimate of the daily averaged all-sky T2m (see discussion, Section 5). Each temperature estimate is associated with three components of uncertainty: random uncertainties on the 0.25 degree daily scale, synoptic scale correlated uncertainty and globally correlated uncertainty excluding uncertainties related to the masking of clouds. The three types of uncertainties are also gathered in a total uncertainty estimate. The land ice temperatures have been calculated for grid cells categorized as ice shelf by ETOPO1, averaged to the 0.25 degree grid (Amante and Eakins, 2009). Sea ice temperatures have been calculated for grid cells with sea ice concentrations above 30-%, according to OSISAF (Tonboe et al., 2016).

[revised manuscript text omitted]

For short-lasting (<24 hours) cloudy conditions the division into 3 h bin averages and the requirement of filled 3 h bins both during night (between 18 and 6 local solar time) and day (between 6 and 18 local solar time) ensure that the diurnal cycle is best resolved despite the gaps with clouds. For long-lasting (>= 24 hours) cloudy conditions $IST_{skin\_L3}$ is not available and we do not retrieve $T2m_{sat}$ for these days. A statistical technique or the use of atmospheric models and assimilation may be used to fill in the gaps. By using a statistical model to combine in situ observed and clear sky satellite derived T2m estimates (over land, lakes, ocean and ice), including uncertainty estimates, EUSTACE has provided a global and gap free daily analysis of surface air temperatures from 1850 to 2015 (Morice et al., 2019).

The product derived here shows an increasing coverage over the time period from 2000-2003 and a stable coverage for 2003-2009. The average daily coverage is 84% and 67% for land ice and sea ice, respectively, considering the stable 2003-2009 period and the 0.25 degree grid. When considering a 1 degree grid resolution, these numbers increase to 94% and 81%, respectively. The high percentages in coverage demonstrate that the gaps due to cloudy days are limited and that the data set contains a significant amount of information on the all-sky daily T2m even though it is based upon clear sky satellite observations.

[revised manuscript text omitted]

5    This model has been used to derive daily T2m on a 0.25 degree regular latitude-longitude grid from the clear sky AASTI $IST_{skin\_L3}$  over the Arctic during the period 2000-2009 (Kennedy et al., 2019), where different regression coefficients have been used for land ice and sea ice. Days with clouds or limited clear sky observations have been excluded from the analysis. Considering a 1 degree regular latitude-longitude grid, the average daily coverage of the $T2m_{sat}$ product is 94% over the GrIS and 81% for sea ice, considering the years 2003-2009. The days when the $T2m_{sat}$ is available

10   the T2m estimate can be considered as a daily averaged all-sky T2m, since it has been tuned against all-sky in situ observations.

[revised manuscript text omitted]

Collins, M., Knutti, R., Arblaster, J., Dufresne, J.-L., Fichefet, T., Friedlingstein, P., Gao, X., Gutowski, W. J., Johns, T., Krinner, G., Shongwe, M., Tebaldi, C., Weaver, A. J. and Wehner, M.: Long-term Climate Change: Projections,

Commitments and Irreversibility, in Climate Change 2013: The Physical Science Basis. Contribution of Working Group I to the Fifth Assessment Report of the Intergovernmental Panel on Climate Change, edited by T. F. Stocker, D. Qin, G.-K. Plattner, M. Tignor, S. K. Allen, J. Boschung, A. Nauels, Y. Xia, V. Bex, and P. M. Midgley, pp. 1029–1136, Cambridge University Press, Cambridge, United Kingdom and New York, NY, USA., 2013.

5  Copernicus Climate Change Service (C3S): ERA5: Fifth generation of ECMWF atmospheric reanalyses of the global climate. Copernicus Climate Change Service Climate Data Store (CDS), [online] Available from: https://cds.climate.copernicus.eu/cdsapp#!/home (Accessed 9 May 2019), 2017.

Dee, D. P., Uppala, S. M., Simmons, A. J., Berrisford, P., Poli, P., Kobayashi, S., Andrae, U., Balmaseda, M. A., Balsamo, G., Bauer, P., Bechtold, P., Beljaars, A. C. M., van de Berg, L., Bidlot, J., Bormann, N., Delsol, C., Dragani, R., Fuentes, M.,
10  Geer, A. J., Haimberger, L., Healy, S. B., Hersbach, H., Hólm, E. V., Isaksen, L., Kållberg, P., Köhler, M., Matricardi, M., McNally, A. P., Monge-Sanz, B. M., Morcrette, J.-J., Park, B.-K., Peubey, C., de Rosnay, P., Tavolato, C., Thépaut, J.-N. and Vitart, F.: The ERA-Interim reanalysis: configuration and performance of the data assimilation system, Q. J. R. Meteorol. Soc., 137(656), 553–597, doi:10.1002/qj.828, 2011.

DuVivier, A. K. and Cassano, J. J.: Evaluation of WRF Model Resolution on Simulated Mesoscale Winds and Surface
15  Fluxes near Greenland, Mon. Weather Rev., 141(3), 941–963, doi:10.1175/MWR-D-12-00091.1, 2013.

Dybbroe, A., Karlsson, K.-G. and Thoss, A.: NWCSAF AVHRR Cloud Detection and Analysis Using Dynamic Thresholds and Radiative Transfer Modeling. Part I: Algorithm Description, J. Appl. Meteorol., 44(1), 39–54, doi:10.1175/JAM-2188.1, 2005a.

Dybbroe, A., Karlsson, K.-G. and Thoss, A.: NWCSAF AVHRR Cloud Detection and Analysis Using Dynamic Thresholds
20  and Radiative Transfer Modeling. Part II: Tuning and Validation, J. Appl. Meteorol., 44(1), 55–71, doi:10.1175/JAM-2189.1, 2005b.

[revised manuscript text omitted]